# Periodicity in the BrO/SO$_2$ molar ratios in the volcanic gas plume of Cotopaxi and its correlation with the Earth tides during the eruption in 2015

Florian Dinger[1,2], Nicole Bobrowski[1,2], Simon Warnach[1,2], Stefan Bredemeyer[3], Silvana Hidalgo[4], Santiago Arellano[5], Bo Galle[5], Ulrich Platt[1,2], and Thomas Wagner[1]

[1]Max-Planck Institut for Chemistry, Mainz, Germany
[2]Institute for Environmental Physics, University of Heidelberg, Germany
[3]GEOMAR, Kiel, Germany
[4]Instituto Geofísico, Escuela Politécnica Nacional, Quito, Ecuador
[5]Department of Space, Earth and Environment, Chalmers University of Technology, Gothenburg, Sweden

*Correspondence to:* Florian Dinger (florian.dinger@mpic.de)

**Abstract.** We evaluated NOVAC (Network for Observation of Volcanic and Atmospheric Change) gas emission data from the 2015 eruption of Cotopaxi volcano (Ecuador) for BrO/SO$_2$ molar ratios. The BrO/SO$_2$ molar ratios were very small prior to the phreatomagmatic explosions in August 2015, significantly higher after the explosions, and continuously increasing until the end of the unrest period in December 2015. These observations together with similar findings in previous studies at other volcanoes (Mt. Etna, Nevado del Ruiz, Tungurahua) suggest a possible link between a drop in BrO/SO$_2$ and a future explosion. In addition, the observed relatively high BrO/SO$_2$ molar ratios after December 2015 imply that bromine degassed predominately after sulphur from the magmatic melt. Furthermore, statistical analysis of the data revealed a conspicuous periodic pattern with a periodicity of about two weeks in a three month time series. While the time series is too short to rule out a chance recurrence of transient geological or meteorological events as a possible origin for the periodic signal, we nevertheless took this observation as a motivation to examine the influence of natural forcings with periodicities of around two weeks on volcanic gas emissions. One strong aspirant with such a periodicity are the Earth tides, which are thus central in this study. We present the BrO/SO$_2$ data, analyse the reliability of the periodic signal, discuss a possible meteorological or eruption-induced origin of this signal, and compare the signal with the theoretical ground surface displacement pattern caused by the Earth tides. Central result is the observation of a significant correlation between the BrO/SO$_2$ molar ratios with the North-South and vertical components of the calculated tide-induced surface displacement with correlation coefficients of 47 % and 36 %, respectively. From all other investigated parameters, only the correlation between the BrO/SO$_2$ molar ratios and the relative humidity in the local atmosphere resulted in a comparable correlation coefficient of about 33 %.

## 1   Introduction

Magmatic melts are a reservoir of dissolved volatile species. Typically, the dominant part of those volatiles is water vapour followed by carbon dioxide and sulphur compounds but also a large number of trace gases like halogen compounds. The

solubility of volatiles in magmatic melts is primarily pressure dependent with secondary dependencies on temperature, melt composition, and volatile speciation (Gonnermann and Manga, 2013). Monitoring the magnitude and composition of volcanic gas emissions has the potential to give insight in those geological processes which vary the solubility regime within a volcanic system. Because the pressure-dependency of the volatile solubility is different for each particular volatile species, molar ratios

of major gas constituents in volcanic plumes are discussed as accessible and suitable proxies for the magma pressure and degassing depth. In particular carbon to sulphur (Burton et al., 2007) and halogen to sulphur ratios, such as chlorine to sulphur (Edmonds et al., 2001) and bromine to sulphur (Bobrowski and Giuffrida, 2012) ratios turned out to be powerful tools enabling to detect events of magma influx at depth, and respectively the arrival of magma in shallow zones of the magmatic system.

The development of passive remote sensing techniques such as Differential Optical Absorption Spectroscopy (DOAS, Platt

et al., 1980; Platt and Stutz, 2008) allows for recording semi-continuous (only during daytime) long-term time series of the magnitudes of volcanic gas emissions as well as analyses of the shape and transport of volcanic gas plumes. Since the early 2000s, the costs of performing such measurements have been drastically reduced due to the installation of automatic remote sensing networks at an increasing number of volcanoes, e.g. Soufriere Hills (Edmonds et al., 2003), Stromboli and Mt. Etna (FLAME network Burton et al., 2009; Salerno et al., 2009), Kilauea (FLYSPEC, Businger et al., 2015), or White Island

(Miller et al., 2006). Most prominently, the Network for Observation of Volcanic and Atmospheric Change (NOVAC, Galle et al., 2010), funded by the European Union in 2005, encompasses today about 100 automatically measuring optical UV-spectrometers at 41 volcanoes which are predominately located in South and Central America. The NOVAC data are recorded by UV-spectrometers which scan across the sky from horizon to horizon in steps of $3.6°$ by means of a small field of view telescope yielding a mean temporal resolution of about 10 min per scan. The usable wavelength of the recorded spectra range

from 280 nm to 360 nm, which allows - in its current state - for an automatised retrieval of the volcanic trace gas concentrations of $SO_2$ (Galle et al., 2010) and BrO (Lübcke et al., 2014) in the volcanic gas plumes.

The utilisation of $BrO/SO_2$ molar ratios as a proxy for volcanic activity has several benefits but also drawbacks in terms of data uncertainty and error sources, if compared to $SO_2$ emission fluxes. First of all, the accuracy of $SO_2$ emission flux estimates may experience high uncertainties due to the atmospheric radiative transport (e.g., Mori et al., 2006; Kern et al., 2010), relies on

the quality of wind data used to approximate plume speed, and requires additional information on plume altitude. In contrast, the $BrO/SO_2$ molar ratios are at most indirectly dependent on the plume speed and plume position, and the radiative transport effects approximately cancel out (Lübcke et al., 2014). On the other hand, while $SO_2$ is directly emitted from the volcano and is assumed to have a lifetime of hours to days (e.g., Beirle et al., 2014; Fioletov et al., 2015), volcanic bromine is considered to be predominately emitted as hydrogen bromine (HBr) and gets only partially converted by photochemistry to BrO by the

so-called bromine explosion process (e.g., Platt and Lehrer, 1997; Wennberg, 1999; von Glasow, 2010). The equilibrium of the conversion HBr $\rightleftharpoons$ BrO is typically reached already within the first minutes after the release of HBr to the atmosphere (Platt and Bobrowski, 2015; Gliß et al., 2015). The NOVAC-instruments (typically installed $5 - 10$ km downwind of the crater and thus observing plumes with an age of about $10 - 30$ min) scan through volcanic gas plumes which are expected to have already reached the BrO equilibrium. The conversion rate or even the stationary state ratio between HBr and BrO may depend

on plume composition and atmospheric conditions (meteorology) (Roberts et al., 2014) and thus on parameters such as solar

irradiance, air temperature, air pressure, and relative humidity. However, Bobrowski and Giuffrida (2012) have not found a correlation between BrO and the relative humidity or the wind speed, which poses the question if the bromine explosion in volcanic gas plumes is indeed significantly depending on the meteorological conditions or not.

The large data set of more than 10 years of semi-continuous time series of volcanic $SO_2$ emission fluxes as well as time series of the $BrO/SO_2$ molar ratios allow for a statistical investigation of long-term signals in the volcanic degassing and comparisons with other long-term signals or a comparison between different volcanoes. The Earth tidal forcing might cause such a long-term signal in the volcanic degassing data. First studies from Stoiber et al. (1986) and Connor et al. (1988) hypothesised about a possible tidal impact on the observed $SO_2$ emission fluxes at Masaya and Kilauea, respectively. The NOVAC data allow to reinvestigate this hypothesis with a much larger data base. Conde et al. (2014) observed a periodic pattern of roughly 13 days in the NOVAC $SO_2$ emission fluxes from Turrialba which was suggested by the authors to be caused by a pressure change due to the Earth tidal forcing. Bredemeyer and Hansteen (2014) used NOVAC data from Villarrica and Llaima to investigate a possible correlation between their $SO_2$ emission fluxes and the surface displacement due to the Earth tides and reported correlation coefficients of up to 20 % and 30 %, respectively. In this manuscript we retrieve the variation of the $BrO/SO_2$ molar ratios recorded by NOVAC during the 2015 period of unrest at Cotopaxi volcano (e.g., Bernard et al., 2016; Gaunt et al., 2016; Morales Rivera et al., 2017) and examine their correlation with variations of local meteorological conditions and partial components of synthetic ground surface displacements obtained from an Earth-tide model.

## 2 Cotopaxi

### 2.1 Geology and volcanism at Cotopaxi

Cotopaxi (-0.7°S, 78.4°W, 5897 m a.s.l.) is a glacier-capped stratovolcano located in the Ecuadorian inter-Andean valley, a N-S to NNE-SSW trending topographical depression. Volcanism in this area is caused by the oblique subduction of the Nazca plate beneath the South American plate, which experiences intense large-scale deformation leading to shortening of the lithosphere along the Andean orogeny (e.g., Trenkamp et al., 2002; Alvarado et al., 2016). The volcano lies on a fault transfer zone (e.g., Gibbs, 1990) in which E-W striking normal faults and NNE-SSW-striking vertical, right-lateral oblique strike-slip faults accommodate the differential crustal shortening rates encountered North and South of Cotopaxi (Fiorini and Tibaldi, 2012). Fiorini and Tibaldi (2012) thus proposed that the fault planes and restraining bends of this transfer zone act as the main magma pathways feeding the volcano. Observations of ground deformation and hypocenter distributions of volcanic earthquakes made in 2001/2002 (Hickey et al., 2015) and 2015 (Morales Rivera et al., 2017) suggest that Cotopaxi currently has a shallow magma reservoir beneath the southwestern flank, which is located at a depth of approximately 5-12 km below the summit. Within the last 0.5 Ma, a rhyolitic and andesitic bimodal magmatism has occurred at Cotopaxi (Hall and Mothes, 2008). The Smithonian Institution (http://volcano.si.edu/) lists 55 eruptions at Cotopaxi within the last 300 years, with the last confirmed eruption in 1940 (VEI 2) or, an unconfirmed eruption in 1942 (VEI 3), and the last major (VEI 4) eruption in 1877. The major hazard of an eruption from Cotopaxi consists in the generation of lahars as a consequence of melting the snow and ice of the summit glacier (Barberi et al., 1995). During the 1877 eruption the glacier of Cotopaxi has completely been molten by the expelled

pyroclastic flows, producing devastating lahars which traveled down the flanks of the volcano and destroyed e.g. the town of Latacunga located about 30 km South of the volcano (Aguilera et al., 2004). Later activity of Cotopaxi did not involve melting of the glacier, which thus recovered since then.

## 2.2 Predicted Earth tidal surface displacement at Cotopaxi location

5  The tidal forces periodically displace the Earth's surface and interior, a summary of the theory of the Earth tides can be found in Appendix A. The periodicities and magnitudes of the tide-induced surface displacement patterns at Cotopaxi are plotted in Figure 1 (predicted with the GNU R software using the package from Gama and Milbert, 2015). The vertical surface

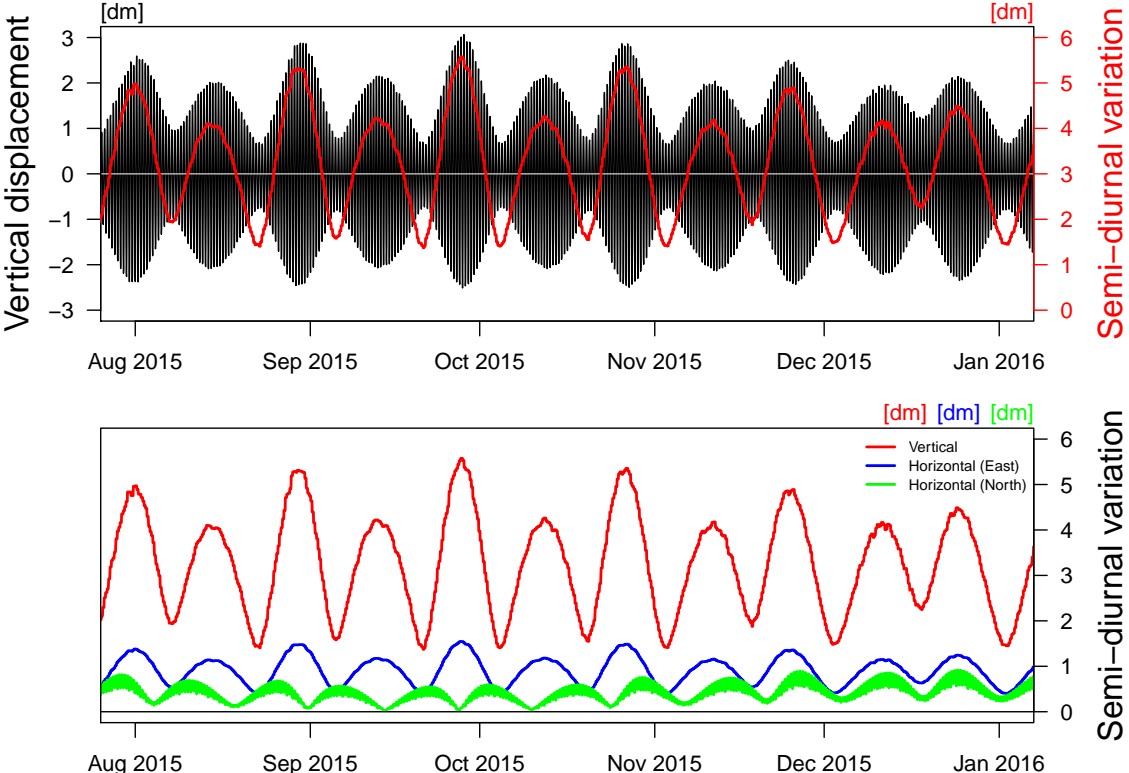

**Figure 1.** Time series of the vertical (top panel and red line in bottom panel) and horizontal (blue and green lines of the lower panel) components of the ground surface displacement at Cotopaxi due to the Earth tides. Upper panel: Hourly resolved vertical surface displacements (black, left ordinate axis) and its daily peak-to-peak variation (red, right ordinate axis). Lower panel: Time series of the daily peak-to-peak variation for all three spatial directions. Vertical and East-West components of the tide are in phase and have a periodicity of 14.8 days. The North-South component of the tide has no strictly regular periodicity but reaches a maximum roughly every 13-14 days and is increasing from September to December 2015.

displacements vary by up to $\pm 0.3$ m within a day and follow a semi-diurnal periodicity (Figure 1, hourly-resolved data in the

upper panel). The semi-diurnal peak-to-peak variation (calculated as the maximum difference within a sliding window of 13 h) of the vertical surface displacements varies between $0.15 - 0.55$ m with a maximum every 14.8 days (Figure 1, red line in upper panel). For ocean tides, this long-term periodicity is called spring-neap tidal cycle, a labelling which we adopt in this paper also for the Earth tides, i.e. maximum daily variations during a spring tide and minimum daily variations during a neap tide.

Notably, each second spring tidal maximum is slightly more pronounced, because the tide-inducing forces of Moon and Sun are slightly stronger during full moon, than they are during new moon. The Earth tides also result in horizontal displacements (see Figure 1 lower panel). Near the equator, where Cotopaxi is located, the East-West displacement is a factor of 5 and the North-South displacement is a factor of 10 smaller than the vertical displacement but for higher latitudes these ratios differ. The East-West component of the tide is in phase with the vertical component (see also Figure 5) while the North-South component

has no strictly regular periodicity but reaches a maximum roughly every 13-14 days. This is mainly due to the elliptic shape of the Moon's orbit and the inclination of the Earth with respect to the ecliptic. Furthermore, due to the variation in the solar declination all tidal components display a semi-annual modulation of the mean displacement with maximum (for the vertical and East-West component) or minimum (for the North-South component) variation at the autumnal and vernal equinox (around September 23 and March 20, respectively).

## 2.3 Cotopaxi's recent unrest and eruption

Around early April 2015 the seismic event rate started to increase and that marked the beginning of a new period of unrest. From mid May 2015 on, also the observed volcanic degassing of $SO_2$ and BrO increased to significant levels above the detection limit. On August 14 2015 several phreatomagmatic explosions opened the conduit resulting in continuous ash emissions and large gas emissions. The period of unrest and eruptions was monitored by the broad network of different sensors maintained

by the local observatory IGEPN in Quito which published the accumulated data as well as interpretations in scientific papers. Those publications discussed the comparison of the temporal variations of magnitude (Bernard et al., 2016) and composition (Gaunt et al., 2016) of ash emissions as well as ground deformation (Morales Rivera et al., 2017) with seismic signals, $SO_2$ emission fluxes, and general visual observations. Morales Rivera et al. (2017) reported a maximum uplift of 3.4 cm at the western flank of Cotopaxi from April to August 2015 while they observed no further significant deformation after the eruption

in August 2015. The authors explain the deformation by an inclined sheet intrusion located a few km southwest of the summit. Bernard et al. (2016) and Gaunt et al. (2016) agree regarding the division of the post-explosion dynamics into four phases: August 14-15 (phase 1), August 15 to October 2 (phase 2), October 2 to November 4 (phase 3), and November 4-11 (phase 4). In phase 1, several phreatomagmatic explosions occurred which generated ash clouds reaching altitudes of 9.3 km above the crater and strong gas emissions. During phase 2, continuous but decreasing ash emissions were observed. In the first half of

phase 2, the number of seismic long-period events as well as the amplitude of tremor was the highest of the overall period of unrest. In Phase 3, tremor and magnitude of ash emissions increased again and peaked in mid of October. In phase 4, only little ash was emitted and the ash plumes did not exceed 2.5 km above the crater.

Gaunt et al. (2016) suggest an intrusion of juvenile magma in a shallow reservoir (less then 3 km depth) since April 2015 as the reason for the reawakening of Cotopaxi, a suggestion backed by the findings of Morales Rivera et al. (2017). The juvenile

magma may have started to heat an overlying hydrothermal system which may ultimately resulted in dry degassing pathways, an interpretation which fits to the observation of shallow $SO_2$ degassing from May 2015 on. The phreatomagmatic explosions removed the plug which consisted of old, altered material. After the explosions, the magma ascent slowed down and/or the residence times at shallow levels increased. As the crystallinity increased, the magma stiffened and a new shallow plug was formed. The authors further highlighted that the fragmentation mechanisms during phase 2 to 4 are not fully understood. Especially the continuous fine ash grain size distribution throughout all phases suggest that several fragmentations took place. As possible interpretation, the authors suggest a repetitive formation and destruction of shallow plugs.

## 3 DOAS measurements of $SO_2$ and BrO

### 3.1 NOVAC stations at Cotopaxi

Today four NOVAC stations are installed around Cotopaxi at distances of 2 to 15 km, predominately downwind (Figure 2). The stations Refugio (instrument no.: D2J2160) and NASA (I2J4969) have operated since 2008. In August 2015, the instrument I2J4969 was relocated from NASA station to the more distantly located new San Joaquin station because the former became unsuitable due to increased ash precipitation. In October 2015, the stations Refugio Sur (D2J2815) and Cami (D2J2835) were additionally installed.

### 3.2 $SO_2$ and BrO time series during the Cotopaxi 2015 period of unrest

We used Differential Optical Absorption Spectroscopy (DOAS) in order to retrieve $SO_2$ and BrO gas emissions from NOVAC data taken during the Cotopaxi period of unrest and eruption (May 2015 - April 2016). Our evaluation is based on Lübcke et al. (2014) but with a modified plume detection algorithm and the option to switch the $SO_2$ fit range when extreme $SO_2$ emissions are observed (see for both issues Appendix B). The output of the evaluation are time series of the slant column densities (SCD) of $SO_2$ and BrO as well as time series of the daily averages of the $BrO/SO_2$ molar ratio in the centre of the volcanic gas plume. $SO_2$-SCDs are the raw data which pose, combined with information of the plume expansion and the wind conditions, the data base for $SO_2$ emission flux estimations. Accordingly, under stable meteorological conditions (similar plume extension and wind speed) $SO_2$-SCDs and $SO_2$ fluxes are highly correlated and share similar long-term behaviour.

From August 2015 to February 2016, Cotopaxi also emitted large amounts of ash. These ash emissions can alter the atmospheric radiative transport and thus result in an underestimation of the retrieved SCDs (e.g., Mori et al., 2006; Kern et al., 2010). Nevertheless, those underestimations are approximately the same for both gas species thus their ratio is almost not affected by those ash emissions (Lübcke et al., 2014). Further, it is under debate whether ash has a differential scavenging effect on sulphur and halogens, respectively (Bagnato et al., 2013; Delmelle et al., 2014).

Prior to May 2015, Cotopaxi was limited to fumarolic degassing and no significant gas plume was detected since the installation of the NOVAC instruments in 2008 ($SO_2$-SCDs $\leq 2 \cdot 10^{17} \frac{molec}{cm^2}$ and BrO-SCDs were below the detection limit of $3 \cdot 10^{13} \frac{molec}{cm^2}$, i.e. twice the standard deviation, first and second panel in Figure 3). Between mid May 2015 and the phreatomagmatic explo-

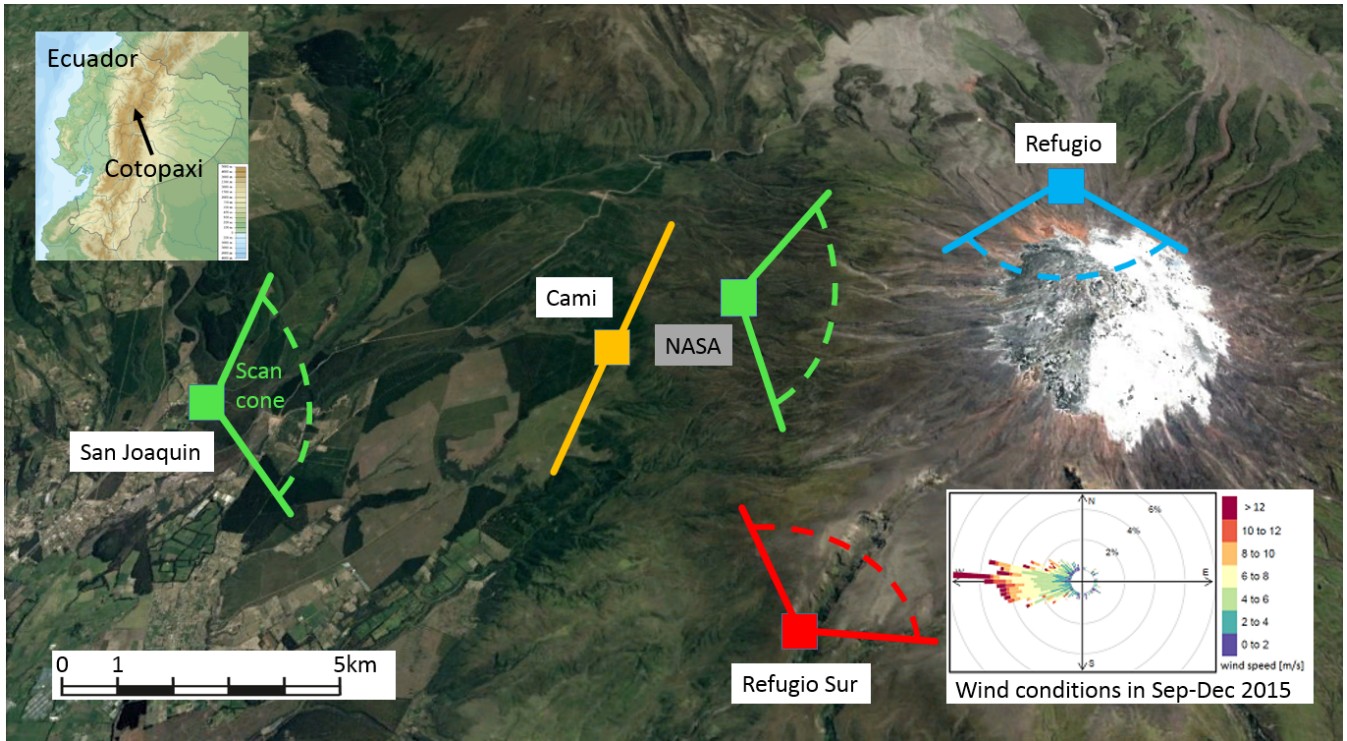

**Figure 2.** Location and scan geometries (see details in Galle et al. (2010)) of the four NOVAC stations at Cotopaxi volcano, the colours match to the time series in Figure 3. After the phreatomagmatic explosions on August 14 2015 the instrument I2J4969 was relocated from NASA station to San Joaquin station. The wind rose in the right bottom corner illustrates the typical wind conditions at an altitude of $100\,\mathrm{m}$ above the crater, i.e. $6000\,\mathrm{m}$ a.s.l. (Map was created with graphical material from Google Earth and wikipedia.com. Wind data were taken from a ECMWF model, see section 4.)

**Table 1.** Typical $SO_2$-SCDs and BrO-SCDs in the gas plume of Cotopaxi in the year 2015. Except for the first time period, the mean values refer to the data which are filtered for the strong plume threshold $SO_2$-SCD $> 7 \cdot 10^{17} \frac{\text{molec}}{\text{cm}^2}$.

| Time period (in year 2015) | mean $SO_2$-SCD | max $SO_2$-SCD | mean BrO-SCD | max BrO-SCD | mean BrO/$SO_2$ |
|---|---|---|---|---|---|
| May 1 - May 22 | $1 \cdot 10^{17} \frac{\text{molec}}{\text{cm}^2}$ | $2 \cdot 10^{17} \frac{\text{molec}}{\text{cm}^2}$ | $\approx 0$ | $4 \cdot 10^{13} \frac{\text{molec}}{\text{cm}^2}$ | |
| May 22 - August 14 | $1 \cdot 10^{18} \frac{\text{molec}}{\text{cm}^2}$ | $2 \cdot 10^{18} \frac{\text{molec}}{\text{cm}^2}$ | $2 \cdot 10^{12} \frac{\text{molec}}{\text{cm}^2}$ | $7 \cdot 10^{13} \frac{\text{molec}}{\text{cm}^2}$ | $0.2 \pm 0.1 \cdot 10^{-5}$ |
| August 14 - August 22 | $2 \cdot 10^{18} \frac{\text{molec}}{\text{cm}^2}$ | $3 \cdot 10^{18} \frac{\text{molec}}{\text{cm}^2}$ | $2 \cdot 10^{13} \frac{\text{molec}}{\text{cm}^2}$ | $8 \cdot 10^{13} \frac{\text{molec}}{\text{cm}^2}$ | $1.1 \pm 0.2 \cdot 10^{-5}$ |
| August 22 - September 8 | (data gap) | | | | |
| September 8 - December 5 | $1 \cdot 10^{18} \frac{\text{molec}}{\text{cm}^2}$ | $1 \cdot 10^{19} \frac{\text{molec}}{\text{cm}^2}$ | $2 \cdot 10^{14} \frac{\text{molec}}{\text{cm}^2}$ | $4 \cdot 10^{14} \frac{\text{molec}}{\text{cm}^2}$ | $4.8 \pm 1.8 \cdot 10^{-5}$ |
| December 5 - May 1 2016 | $1 \cdot 10^{18} \frac{\text{molec}}{\text{cm}^2}$ | $2 \cdot 10^{18} \frac{\text{molec}}{\text{cm}^2}$ | $4 \cdot 10^{13} \frac{\text{molec}}{\text{cm}^2}$ | $1 \cdot 10^{14} \frac{\text{molec}}{\text{cm}^2}$ | $4.4 \pm 1.6 \cdot 10^{-5}$ |

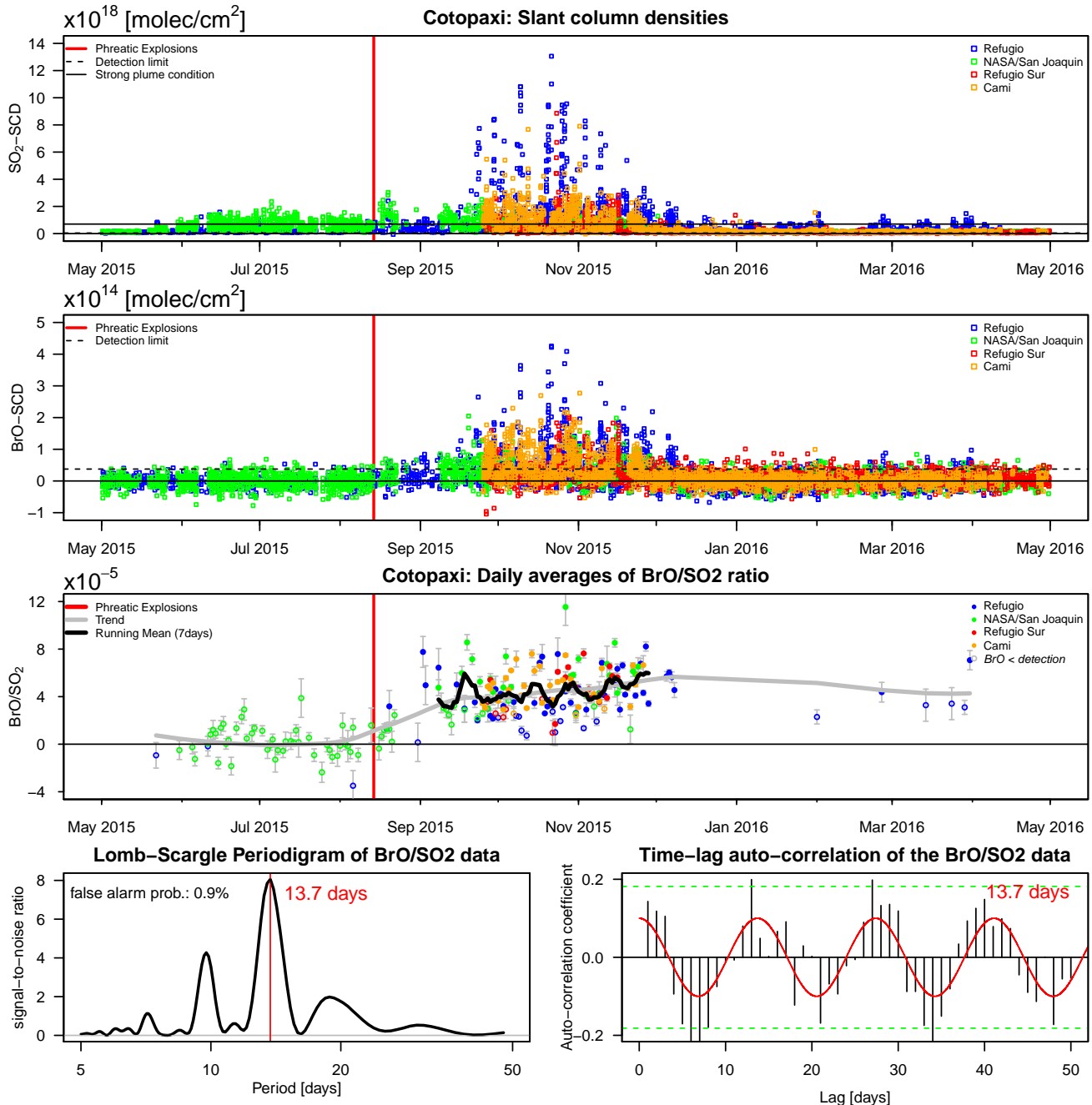

**Figure 3.** Three upper panels: Time series of the slant column densities of $SO_2$ and BrO and calculated $BrO/SO_2$ molar ratios in the gas plume emitted from Cotopaxi (tick marks indicate first day of the particular month). The NOVAC stations are indicated by the different colours. $BrO/SO_2$ molar ratios: The trend (grey line) is calculated by a local regression low pass filter. Ratios obtained from BrO data below the detection limit of two standard deviations are highlighted by open circles (see Appendix B2). Lowest panels: Lomb-Scargle frequency component analysis and time-lag auto-correlation coefficients (dashed lines give 90 % confidence level) of the $BrO/SO_2$ molar ratio variations. Both analyses are based on the trend-corrected data acquired during the time interval indicated by the running mean line in the $BrO/SO_2$ molar ratio plot (see text).

sions on August 14 2015 the SO$_2$-SCDs increased up to $2 \cdot 10^{18} \frac{\text{molec}}{\text{cm}^2}$ and the BrO-SCDs increased up to $7 \cdot 10^{13} \frac{\text{molec}}{\text{cm}^2}$. Within this period, the calculated BrO/SO$_2$ molar ratios were extremely low with an average of $0.16 \pm 0.13 \cdot 10^{-5}$ (third panel in Figure 3). No variations can be resolved because the values are close to the BrO detection limit.

Within the week directly after the phreatomagmatic explosions on August 14 2015, mean SO$_2$-SCDs of $2 \cdot 10^{18} \frac{\text{molec}}{\text{cm}^2}$ were observed with a maximum value of $3 \cdot 10^{18} \frac{\text{molec}}{\text{cm}^2}$, the mean BrO-SCDs increased from approximately 0 to $2 \cdot 10^{13} \frac{\text{molec}}{\text{cm}^2}$, to a maximum of $8 \cdot 10^{13} \frac{\text{molec}}{\text{cm}^2}$, and a mean BrO/SO$_2$ molar ratio of $1.1 \pm 0.2 \cdot 10^{-5}$ was derived. From August 22 to September 8 2015, there are only sparse plume data recorded because the solar panel at NASA station was temporarily covered by ash and the gas plume was apparently not transported through the field of view of the Refugio station during this period. After September 8 2015, a drop in the SO$_2$-SCDs derived from the data measured at the NASA and San Joaquin stations (green dots in Figure 3) is expected independently from a possible variation of the volcanic degassing behaviour. The reason is the relocation of the instrument to a larger distance to the emission source and therefore observing more diluted volcanic gas plumes.

From early September 2015 to mid of December 2015, the mean SO$_2$-SCDs remained approximately on the same level as directly after the explosions (significant amount of data above $1 \cdot 10^{18} \frac{\text{molec}}{\text{cm}^2}$ but typically below $3 \cdot 10^{18} \frac{\text{molec}}{\text{cm}^2}$), however with strong temporal variability (compared to the emission before September 2015) with a maximum of $13 \cdot 10^{18} \frac{\text{molec}}{\text{cm}^2}$. In contrast, much higher BrO-SCDs were observed in this period with daily maxima typically varying between $1 - 2 \cdot 10^{14} \frac{\text{molec}}{\text{cm}^2}$ but up to $4 \cdot 10^{14} \frac{\text{molec}}{\text{cm}^2}$. As expected with respect to the gas plume dilution, an intercomparison of the data obtained from the different stations reveals that the observed SCDs in both SO$_2$ and BrO are larger the closer the station is to the edifice of Cotopaxi, however, all stations observed similar BrO/SO$_2$ molar ratios. Further, Refugio station which is installed just 2 km North of Cotopaxi observed some extreme SCD-values (simultaneously for SO$_2$ and BrO) which are not at all observed by the other stations. A comparison with the wind profile (see also Figure 4) revealed that at least three (on September 29, October 9, October 20) of those seven SO$_2$-peaks (with SO$_2$-SCDs $> 7 \cdot 10^{18} \frac{\text{molec}}{\text{cm}^2}$) coincide with extreme low wind speeds ($< 2 \frac{\text{m}}{\text{s}}$), thus at those days the elevated SCDs may be caused by an accumulation of the volcanic gas emissions in proximity to the crater. Accumulated gas plumes are expected to have a reduced BrO/SO$_2$ molar ratio due to the limited lifetime of BrO. For the three mentioned days, the mean BrO/SO$_2$ molar ratios indeed dropped to $2 \cdot 10^{-5}$, thus despite the observed large BrO-SCDs, we have to be cautious when interpreting the BrO/SO$_2$ molar ratios of those days. Within the period from early September 2015 to early December 2015 a mean BrO/SO$_2$ molar ratio of $4.8 \cdot 10^{-5}$ but with variations within $1 - 12 \cdot 10^{-5}$ was observed. Further, the BrO/SO$_2$ molar ratios increased from early September 2015 on with a trend of about $0.3 \cdot 10^{-5}$ per month.

Since early December 2015 the mean SO$_2$-SCDs dropped to levels comparable to those prior to the explosions, however, with some minor SO$_2$ degassing events which were detected exclusively by the Refugio station. Also the mean BrO-SCDs dropped to $1 \cdot 10^{13} \frac{\text{molec}}{\text{cm}^2}$, i.e. below the detection limit, however, in contrast to the situation prior to the explosions for most days daily maxima above the detection limit were observed with values of even up to $1 \cdot 10^{14} \frac{\text{molec}}{\text{cm}^2}$ on few days. Around March 2016 we observed some strong gas expulsions approaching SO$_2$-SCDs, which exceeded $7 \cdot 10^{17} \frac{\text{molec}}{\text{cm}^2}$. The corresponding BrO/SO$_2$ molar ratios were about $4.4 \cdot 10^{-5}$, thus similar to the post-explosion value in September 2015. Despite that the reliability of this observation is limited due to the small sample size, these data points indicate that the long-term average value of the BrO/SO$_2$ molar ratios was about $4 - 5 \cdot 10^{-5}$ during this period.

### 3.3 Periodic pattern in the BrO/SO$_2$ molar ratios after the explosions

Within the time interval from early September 2015 to early December 2015, the highly significant BrO absorptions and the up to four simultaneously monitoring NOVAC stations yield an extraordinarily good temporal resolution of the BrO/SO$_2$ time series of almost daily resolution. Therefore besides the monotonous long-term trend also "high-frequency" (with periods of weeks or months) variations can be resolved in the BrO/SO$_2$ data. And indeed, the maxima and minima of the BrO/SO$_2$ molar ratios apparently follow a periodic pattern with a period of about two weeks and an amplitude (of the smoothed data) of about $1 \cdot 10^{-5}$, i.e. a relative variation of about 20 %, as illustrated by the running mean over 7 days (thick black line in third panel in Figure 3).

In order to examine a quantitative estimation for the reliability of this signal we tested two statistical methods, a Lomb-Scargle frequency component analysis (Lomb, 1976; Scargle, 1982; Press et al., 1992, in the following just called Lomb-Scargle analysis) as well as a temporal auto-correlation analysis. Compared with a common Fourier transform analysis those two methods can more appropriately handle time series with irregular sampling intervals, a problem we are facing as long as we are not interpolating the BrO/SO$_2$ molar ratio time series at the missing days. Namely, the Lomb-Scargle analysis is similar to a Fourier transform analysis but explicitly designed for the examination of irregularly spaced time series and while the auto-correlation analysis still requires a linear interpolation of the BrO/SO$_2$ time series the distortion on the correlation coefficients is much weaker than for a Fourier transform analysis based on sinusoidal signals. Both methods were applied to the daily data (arithmetically averaged over all instruments when daily means were available for several instruments) for the time interval from September 8 2015 to December 15 2015 (as illustrated by the black line of the running mean in the third panel of Figure 3). The first week of September 2015 and the first week of December 2015 were excluded from both analyses as the temporal resolution is much worse at those times. Also the trend was removed by subtracting a polynomial fit to the fifth order from the BrO/SO$_2$ time series. (This should however not be confused with the trend shown in the third upper panel of Figure 3, which is just a smooth guidance for the eye. Nevertheless, the numerical values of those both trend fits are comparable for the time interval from September to December 2015.)

The Lomb-Scargle analysis identifies a periodicity of 13.7 days and the auto-correlation analysis confirms the existence of a periodicity of 13-14 days (lowest panels in Figure 3). For both methods, the results are highly significant (Lomb-Scargle false alarm probability of 0.9 % and an auto-correlation detection confidence of about 90 %, respectively) and thus the periodic signal is with high probability an observation of a natural process (or a series of those) rather than statistical noise. On lower confidence levels, the Lomb-Scargle analysis identifies further peaks at a periodicity of 7.1 days, 9.7 days, and 18.8 days, respectively.

### 4 Meteorological conditions during the Cotopaxi eruption

The atmospheric chemistry and meteorological conditions may have an impact on the BrO/SO$_2$ molar ratios observed in volcanic gas plumes (see introduction). Unfortunately, direct measurements of the meteorological conditions at volcanoes, and in particular at the altitudes of volcanic plumes, are scarcely or not at all available. A quantification of the sky cloudiness or

the aerosol load in a volcanic plume may be retrieved directly from the spectroscopic data by well-established methods such as an analysis of variations in the oxygen dimer $O_4$ absorption or an analysis of the so-called colour index (Wagner et al., 2014). However, the spectra recorded by NOVAC do not allow for an application of the the validated versions of those methods because the spectral properties are optimised for a retrieval of $SO_2$ and BrO, thus for a wavelength range below 360 nm. In contrast, the $O_4$ analysis requires spectra of good quality also at longer wavelengths. Analyses with modified versions of those methods (see Appendix B4) did not find any reliable pattern in the time series of the sky cloudiness.

Weather forecast models pose the best available proxy for the meteorological conditions around the volcanic gas plume. We use operational analysis data from ECMWF (on a Gaussian Grid N640 and with a T1279 truncation) with a spatial resolution of 0.14° x 0.14° and a temporal resolution of 6 h (0:00, 6:00, 12:00, and 18:00 UTC) and picked the model altitude around 6000 m a.s.l. (100 m above the crater, where we roughly expect the volcanic gas plume) in order to get a rough idea of the meteorological variation at Cotopaxi volcano. From the list of available model parameters, we selected 1) the air temperature, the air pressure, and the relative humidity in order to investigate their possible influence on plume chemistry, 2) the wind speed and wind direction in order to reconstruct the plume transport (and therefore plume age), and 3) the total cloud cover as a poor but still most suitable proxy for the radiative conditions. The total cloud cover gives a relative value (from 0 to 1) for the cloud thickness above a pixel integrated from ground up to an altitude of 60 km. However, a total cloud cover of 1 does not necessarily indicate the presence of an optical dense cloud cover close to the ground, but rather indicates whether significant light scattering within clouds has occurred somewhere above the ground pixel. At the equator, where Cotopaxi is located, the total cloud cover is typically close to 1. For our comparison of gas and meteorological data we chose to focus on the data obtained at 18:00 UTC, i.e. 13:00 local time, which match the observation time of the DOAS measurement the best, especially when we compare with the daily averages of the $BrO/SO_2$ molar ratios which are centred around noon time.

During the period of unrest, some of the meteorological parameters experienced strong but irregular variations (Figure 4). We applied a Lomb-Scargle analysis and an auto-correlation analysis to the de-trended time series, which were obtained by subtracting a polynomial fit of the fifth order from the time series. When applied to the total presented meteorological time series (July 1 2015 to January 1 2016), no persistent periodic patterns were detected. When limiting the analysis to the time interval of the investigated $BrO/SO_2$ data (i.e. from September 8 2015 to December 15 2015), both the variations of relative humidity and cloud cover show noticeable periodic patterns. The relative humidity was at its half-year minimum of 10 % in mid September 2015 and increased to its half-year maximum of 96 % in mid of October 2015/early November 2015. Further, this trend was superimposed by a significant (Lomb-Scargle: false alarm probability of 1.2 %) periodicity of 13.1 days. The total cloud cover was at its half-year minimum of 0.6 in mid September 2015 and increased to its half-year maximum of (persistently) 1.0 early November 2015. Further, for the time interval from September 2015 to December 2015 a periodicity of 11.6 days was proposed (Lomb-Scargle: false alarm probability of 2.2 %). An auto-correlation analysis confirms the existence of the three noted periodicities, all with a confidence of about 90 %. For the other parameters no significant periodic patterns were detected.

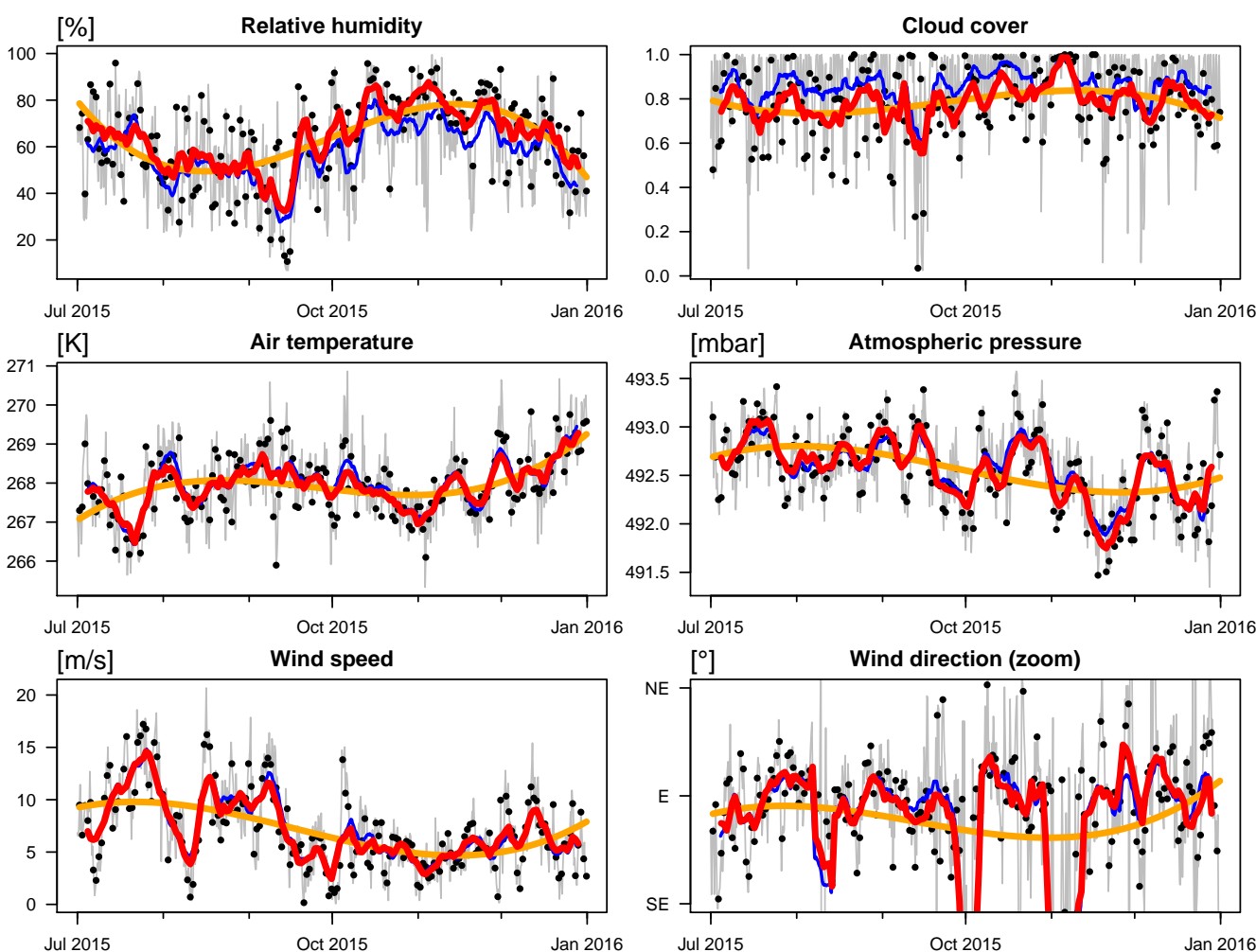

**Figure 4.** Meteorological conditions at 100 m above the peak of Cotopaxi (6000 m a.s.l.) during the period of unrest. The data are based on operational analysis data from ECMWF with a spatial resolution of $0.14° \times 0.14°$. The grey lines show the ECMWF data with a temporal resolution of 6 h, given in UTC, the black dots show the data at 18:00 UTC, i.e. 13:00 local "noon" time. Running means of 7 day windows are shown as blue lines for the total data set and as red lines for the noon time data, which mostly fall on top of the blue line. The overall long-term trend is indicated by the polynomial fit of fifth order shown in orange. For the wind direction, only data with an easterly component are shown as this is the predominately wind direction (93 % of data) and as westerly components exclusively coincide with low wind speeds ($< 4 \frac{m}{s}$).

## 5 Discussion

A variation of the BrO/SO$_2$ molar ratios in the volcanic gas plume could be caused by effects in two independent domains: either within the magmatic system, i.e. varying magmatic processes could cause a variation in the HBr/SO$_2$ molar ratio within the volcanic gas emissions; or in the atmosphere where the atmospheric conditions may delay or accelerate the conversion from HBr to BrO or even change the chemical steady state ratio between HBr (or other bromine compounds) and BrO.

We observed three significant variations in the BrO/SO$_2$ data: (1) an abrupt increase of the molar ratios in September 2015 (unfortunately the exact behaviour is hidden by the data gap), (2) an increasing trend from September 2015 to December 2015, and (3) the periodic pattern from September 2015 to December 2015. In the following we compare the variations of the volcanological, meteorological, and calculated tidal data with the variations of the BrO/SO$_2$ data. For the time interval from September 8 2015 to December 15 2015, we performed a correlation analysis between the BrO/SO$_2$ data, the meteorological data, and the calculated tidal data. We compared the daily averages of the gas data (calculated from data exceeding the strong plume threshold indicated in the upper panel of Figure 3), the noon data of the meteorological data, and the shifted (see below) tidal data. All data sets were left with their long-term trends in order to compare the total variations (see Figure 5). An analogue analysis with trend-corrected data led to similar results but lower correlation coefficients.

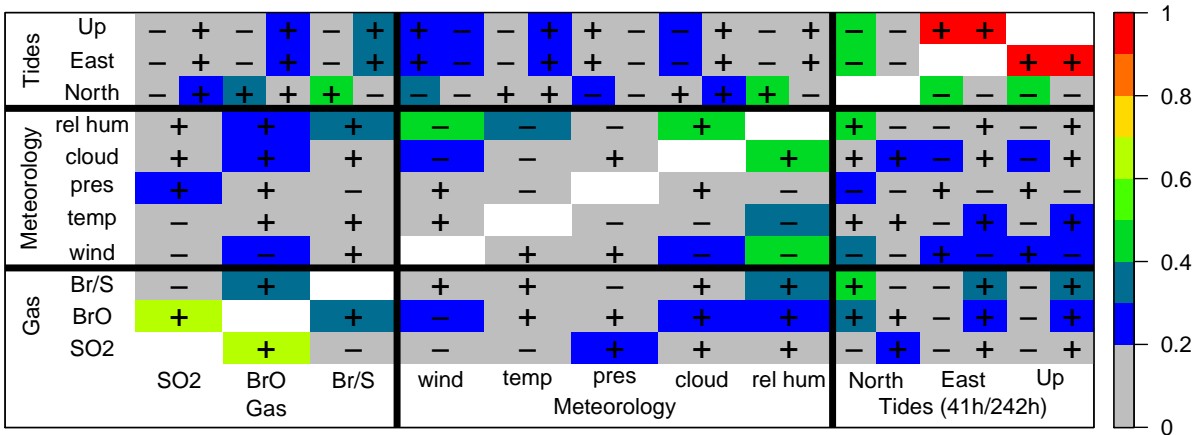

**Figure 5.** Correlation coefficients of the different data sets, where "Br/S" is used as abbreviation for BrO/SO$_2$ molar ratios. The colours indicate the absolute value of the coefficient divided into bins of 10 %. The signs of the correlations are indicated by the "+" and "-", respectively. The unity auto-correlations are omitted for better readability. In order to account for the delay between tidal forcing and atmospheric measurement, the time series of the tides was shifted backwards in time prior to the correlation analysis. The figure shows the correlation for a shift of 41 h (left column) and for a shift of 242 h (right column), see text for derivation of the shifts. For both shifts, the North-South component correlates the strongest with the BrO/SO$_2$ molar ratios.

## 5.1 Comparison of BrO/SO$_2$ time series with gas emission rates and meteorology

The BrO/SO$_2$ molar ratios are not correlated to the daily means of the SO$_2$ slant column density (correlation coefficient of -10 %, see Figure 5) but are partially correlated to the daily means of the BrO slant column densities (39 %). It is interesting to note that similar observations can be made when comparing BrO/SO$_2$ molar ratios and SO$_2$ emission rates obtained from the same instruments. For this purpose a time series of the SO$_2$ emission rates was provided by the group at IGEPN, who have access to a novel wind model with enhanced temporal and spatial resolution allowing for the calculation of more accurate emission rates. A publication of the respective time series of the SO$_2$ emission rates is in preparation. Based on the IGEPN data, BrO/SO$_2$ molar ratios and SO2 emission rates are not correlated during the period of unrest.

When limiting the analysed time interval to early September 2015 to mid December 2015, relative humidity, total cloud cover, air temperature, and wind speed show long-term variations (Figure 4), however, those are superimposed by strong short-term variations. For the relative humidity, a frequency analysis finds a significant high-frequency signal with a 13.1 days periodicity which matches relatively well the 13.7 days periodicity of the BrO/SO$_2$ data. A cross-correlation analysis of the limited time interval finds a slightly increased correlation coefficient of 33 %. For air temperature, air pressure, and wind conditions no significant periodicities were found and the periodic pattern in the total cloud cover (period of 11.6) does not match with the BrO/SO$_2$ variations. As conclusion, the BrO/SO$_2$ variations may be partially explained by the variations in the relative humidity. However, on the other hand the correlation between the relative humidity and the BrO/SO$_2$ molar ratios may just be caused by a shared dependency on, e.g., the Earth tidal forcing (see good correlation between relative humidity and tide-induced surface displacement in Figure 5).

## 5.2 Comparison of the BrO/SO$_2$ time series with the Earth tides

It should be kept in mind that one can expect a delay between a tide-driven process in the volcanic system and the finally observed volcanic gas emissions. We determined the delay by a cross-correlation analysis between the BrO/SO$_2$ molar ratios and the calculated tidal surface displacement. For this purpose, we shifted the time series of the Earth tide data variably by up to two weeks backwards in time and thereby manually maximised the cross-correlation coefficient between BrO/SO$_2$ and tide-induced surface displacement. However, the optimal delay is different when we optimise for the different components of the tidal signal: it is 242 h when we optimise with respect to the vertical component (or East-West component which is perfectly in phase with the vertical component) but 41 h when we optimise for the North-South component. As a result we derived correlation coefficients between the BrO/SO$_2$ molar ratios and the vertical component or the North-South component of 36 % and 47 %, respectively (see Figure 5 where both optimisation scenarios are included). As last analysis step, we dropped the handy representation (radial, North-South, East-West) of the tide-induced surface displacement but correlated the displacement in variable directions (calculated as the component of the displacement vector in the chosen three-dimension direction) with the BrO/SO$_2$ molar ratios. The overall optimum was again found to be strictly horizontal and quite strictly to the North (optimum was 1° NNE but with similarly high correlation coefficients for ±8°) and again with a delay of 41 h.

The tidal North-South displacement component fits extraordinarily well to both, the trend and the "skewed" periodic pattern in

the BrO/SO$_2$ data (left panel in Figure 6). Further, a scatter plot revealed a highly significant contrast in BrO/SO$_2$ data between small and large semi-diurnal tidal deviations (right panel in Figure 6).

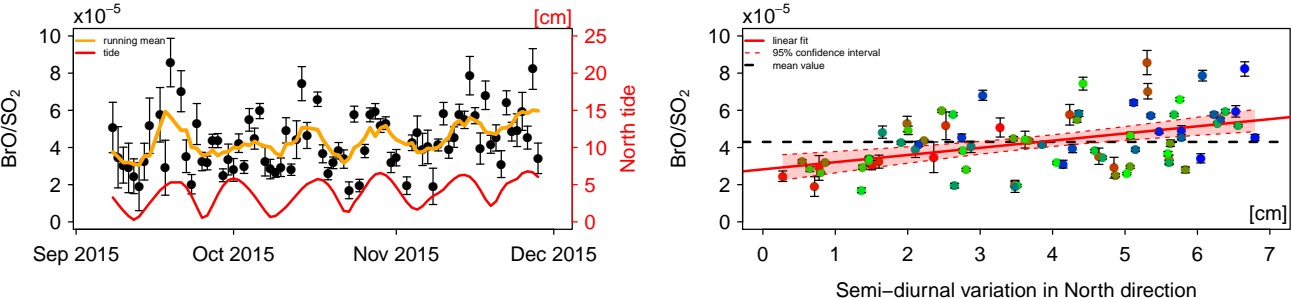

**Figure 6.** Graphical comparison of temporal variations of the BrO/SO$_2$ molar ratio and the predicted tide-induced surface displacement variation in North-South direction. Left panel: Comparison of the time series. Right panel: scatter plot which illustrates the correlation coefficient of 47 % between both time series, the gradual changing colour of the data points gives the measurement date from red (September 8 2015) to blue (December 15 2015).

## 5.3    Comparison of BrO/SO$_2$ time series with volcanic activity

### 5.3.1    BrO/SO$_2$ molar ratios as a poorly constrained proxy for volcanic activity

The volatile solubility in magmatic melt can be modelled with high accuracy for water vapour and carbon dioxide (Gonnermann and Manga, 2013; Oppenheimer et al., 2014) and is also relatively well constrained under diverse and variable conditions for sulphur (e.g., Scaillet and Pichavant, 2005; Moretti and Ottonello, 2005; Lesne et al., 2011; Witham et al., 2012). In contrast, it is more challenging to constrain the halogen behaviour in the melt-fluid phase because of their low abundance in magmatic melt but also because halogens (except fluorine) strongly prefer to partition in the hydrous fluid phase (Bureau et al., 2000). For

water-poor magmas, the knowledge is limited to the qualitative observation that halogens typically reach saturation, if at all, at low pressure (Edmonds et al., 2009). For water-rich magmas, the quantitative pressure-dependency of the chlorine solubility in the melt-fluid phase is relatively well constrained (e.g., Carroll, 2005; Balcone-Boissard et al., 2016) and also some insights in the behaviour of the other halogens have been gathered by the experimental (e.g., Bureau and Métrich, 2003; Bureau et al., 2010, 2016) or empirical (e.g., Villemant et al., 2005; Kutterolf et al., 2015) derivation of their melt-fluid partition coefficients.

Sulphur and water vapour, and thus also bromine, are predominately degassing after carbon dioxide. Bromine has a larger partition coefficient than chlorine, thus it is reasonable that bromine degasses prior to chlorine. Sulphur is degassing prior to chlorine (e.g., Aiuppa, 2009), however, it is not yet constrained whether sulphur is predominately degassing prior or after bromine. Accordingly, even under the assumption that the HBr/SO$_2$ degassing ratio is purely pressure-dependent no quantitative deduction of the pressure change could be made based on BrO/SO$_2$ molar ratios. Further, it was even not yet possible to

derive the sign of this correlation indirectly by empirical studies of BrO/SO$_2$ molar ratios simultaneously to pressure changes

(Bobrowski and Giuffrida, 2012; Lübcke et al., 2014) because of the still limited data sets available. In the absence of an explicit interpretation of the $BrO/SO_2$ molar ratio, at least the reasoning that a change in $BrO/SO_2$ molar ratios is probably caused by a significant change in magmatic conditions appears to be nevertheless a reasonable hypothesis. Several studies on $BrO/SO_2$ molar ratios in volcanic gas plumes revealed that the ratios generally follow the same typical pattern: the ratios are

relatively low prior and during an eruption but are higher at the end of the eruption or during quiescent degassing periods, as e.g. observed at Mt. Etna (Bobrowski and Giuffrida, 2012), Nevado del Ruiz (Lübcke et al., 2014), Tungurahua (Warnach, 2015), and also here for Cotopaxi.

### 5.3.2  Comparison of $BrO/SO_2$ molar ratios with volcanological data

During the week after the explosions on August 14 2015, the $BrO/SO_2$ molar ratios remained on a similarly low level

($\leq 1 \cdot 10^{-5}$) as prior to the explosions. The $BrO/SO_2$ abruptly increased to a mean value of about $4 \cdot 10^{-5}$ during the next two subsequent weeks. Gaunt et al. (2016) mentioned that the hydrothermal system might have been drying during those weeks. Accordingly, at first bromine predominately partitioned in the hydrous fluid of the hydrothermal system, but after the hydrothermal system has been dried out the bromine has probably directly partitioned to the gas phase.

The increasing trend in $BrO/SO_2$ from mid of September 2015 to early December 2015 indicates a slow but continuous change

in magma pressure, magma temperature, or composition of the degassing magma body. The possible reasons are manifold. Following Gaunt et al. (2016), the mingling of the old magma with juvenile magma could have continuously changed the effective magma composition. Or, in the case that HBr degasses at higher pressures than $SO_2$, a gradual pressure increase due to e.g. the formation of a new plug may have resulted in suppressed $SO_2$ degassing. If in contrast HBr is degassing at lower pressures than $SO_2$, the increasing trend may indicate that the magma has already degassed most $SO_2$ while the magma still held significant HBr which yet had to degas. The latter interpretation would be supported by the relatively large BrO-SCDs but

rather small $SO_2$-SCDs since January 2016.

### 5.3.3  Possible volcanological origins for the periodic pattern in the $BrO/SO_2$ molar ratios

The volcanic unrest and in particular the explosions in August 2015 mark a fundamental perturbation of the equilibrium conditions in the volcanic system. Thus, the observed periodic pattern in the $BrO/SO_2$ data might be a manifestation of a

damped and oscillating restoring process of the volcanic system to a new equilibrium. However, the damping of magma motion is rather high and thus a one-time excitement of the magma column, e.g. by the phreatomagmatic explosions in August 2015, is unlikely to cause an observable oscillation of six consecutive periods. As another possible explanation, the periodic pattern may have been caused by a chance repetition of six one-time events at equidistant time intervals, e.g. by a repetitive plug formation and destruction as suggested by Gaunt et al. (2016). Anyway, even if the interpretation of a repetitive plug

formation holds true, the probability that those events happened at equidistant timing is expected to be rather low unless the events are triggered by a periodic external forcing. The spring-neap tide cycle of the Earth tides is such a continuous periodic forcing which, at least in principle, is continuously varying the magmatic system. Finally, we can not exclude that the unrest or geometry itself might have led to an accidental periodicity in the activity of two weeks which was also completely independent

from tidal influence, e.g. a periodic magma intrusion from the deep to the shallow reservoir due to tectonics. However, again, it appears to be unlikely that such an accidental periodicity has a period of just two weeks. Thus from all geological origins, the tidal forcing appears to be the best candidate to explain the observed periodic signal in the BrO/SO$_2$ data.

## 5.4   Interpretation of the periodic pattern in the BrO/SO$_2$ data

The observed periodicity in the time series of the BrO/SO$_2$ molar ratios is superimposed by an increasing trend and a large scatter in the data. The latter highlights the complexity of the interpretation of BrO/SO$_2$ molar ratios, which potentially depends on an unknown number of volcanological and atmospherical mechanisms and the fluctuations of their parameters. Despite the large scattering, we nevertheless found an unexpectedly high correlation between the BrO/SO$_2$ molar ratios on the one hand and the Earth tidal forcing or the relative humidity on the other hand, with correlation coefficients of 47 % and 33 %, respec-

tively. Thus, the tidal forcing as well as the relative humidity are the most probable candidates to explain a part of the variability of the BrO/SO$_2$ molar ratios. Accordingly, both mechanisms may independently contribute to the variation of the BrO/SO$_2$ molar ratios at the same time. In the following, we focus on the plausibility of a causality between the BrO/SO$_2$ molar ratios and the North-South component of the tidal forcing, which nevertheless appears to be the best candidate.

Tide-generated pressure and stress variations appear to be negligibly small in comparison to the forces that typically act within magmatic systems (Emter, 1997). However, it has been shown in several cases that the effect of these subtle variations can be amplified and thus may play a crucial role if 1) the system is sufficiently open (McNutt and Beavan, 1984; Sottili and Palladino, 2012), and 2) the geometry of the magma feeding systems and reservoirs allows for a tide-induced periodic deformation (e.g. Patanè et al., 1994; Sottili et al., 2007), which additionally may be reflected by a periodic spatiotemporal cumulation of

volcanic earthquakes in the excited parts of the system (Rydelek et al., 1988; McNutt, 1999; Custodio et al., 2003). Sottili and Palladino (2012) e.g. attributed the cyclic clustering of eruptive activity at Stromboli volcano and its concurrence with the times of the syzygies to the tidal response of well-known tectonic structures causing a periodic magma supply and migration in the shallow parts of the plumbing system.

In this reasoning, the shallow emplacement of magma prior to the phreatomagmatic explosion in August 2015 may have

(temporarily?) shifted the geometry of the Cotopaxi magma plumbing system from a non-excitable to an excitable state. Additionally, the explosive activity very likely gave rise to a transition from closed system to open system degassing, and thus made the volcano more perceptive to external influences. Further, combining our results and the interpretation from Gaunt et al. (2016) suggests a possible tide-induced repetitive plug formation and destruction, causing an alternation of "open and almost shut condition" as proposed by Fischer et al. (2002) for explosive activity at Karymsky volcano. All those effects can result

in a periodic variation of the pressure regime in the shallow magmatic system, leading to a periodic variation of the volatile solubility in the magmatic melt, which in turn may vary the magnitude and/or composition of volcanic gas emissions.

Tide-induced processes are intuitively expected to strictly follow the periodicity of the strongest tidal long-term pattern, that is the spring-neap tide cycle with period of 14.8 days. The observed periodic pattern of 13.7 days in our BrO/SO$_2$ data, however,

matches much better with the temporal intraday amplitude variation of the North-South component of the tide-induced surface displacement, which follows a rather irregular pattern with maxima occurring roughly every 13-14 days. In other words, BrO/SO$_2$ molar ratios were elevated when the tidal amplitude variations in North-South direction were most pronounced. Our results accordingly suggest that the volcanic system of Cotopaxi (currently?) is more sensitive to tide-induced stresses acting in the North-South direction, rather than to stresses in the vertical and East-West directions.

This interpretation is further supported by the orientation of the local fault system and associated ambient stress field at Cotopaxi, which is located in a transfer fault zone with greatest principal stress acting in the ENE-WSW direction and the weakest principal stress in the North-South, i.e. also in horizontal direction (Fiorini and Tibaldi, 2012). Such a setting favours the intrusion and ascent of magma along East-West striking planar structures, which is further corroborated by the observation of the inclined sheet intrusion beneath the south-western flank of the volcano (Morales Rivera et al., 2017). Thus, the additional tide-generated stresses probably have a much higher relative impact, when they act in the direction of weakest principle stress, i.e. normal to East-West striking compressible magma pathways, if compared to the other directions. Such a directional dependency is indeed well known for the tidal response of inclined planar aquifers, which cross-cut borehole wells. Bower (1983) e.g. reported oscillations of water levels in boreholes in Canada, which indicated a strong response to the horizontal component of the semi-diurnal M2 tide acting normal to the strike direction of the intersecting aquifers.

## 6   Conclusions

Previous studies on the volcanic gas plumes of several volcanoes (Mt. Etna, Nevado del Ruiz, Tungurahua) observed relatively low BrO/SO$_2$ molar ratios prior to volcanic explosions and an increasing trend in BrO/SO$_2$ molar ratios afterwards. Those consistent observations raised the question whether the BrO/SO$_2$ molar ratios can be interpreted as a precursor of volcanic activity, and to which extent these can serve as an indicator for pressure fluctuations in the degassing magma residing at shallow levels of the magma plumbing system. We observed a similar behaviour at Cotopaxi during its unrest period in 2015, extending the empirical foundation of this claim. At Cotopaxi, the BrO/SO$_2$ molar ratios were almost vanishing prior to the phreatomagmatic explosions in August 2015, significantly higher after the explosions, and further increased from September 2015 to December 2015. After December 2015, the unrest calmed down accompanied by a decrease in SO$_2$-SCDs to a level lower than prior to the explosions, however, the BrO-SCDs remained relatively large. The latter observation suggests that bromine degassed at Cotopaxi predominately after sulphur from the magmatic melt.

Furthermore, we observed a periodic pattern with a period of 13.7 days in the BrO/SO$_2$ molar ratios within the volcanic gas plume of Cotopaxi. Despite the fact that the analysed time interval spans just over six periods of this signal, the statistical significance is unexpectedly high in the view of the manifold processes that may potentially influence the molar gas ratio (false alarm probability of 0.9 % and detection confidence of about 90 %). At first, we examined the variability of the meteorological conditions as origin for the periodic variation of the BrO/SO$_2$ molar ratios. We found that the relative humidity exhibits a similar long-term trend as the BrO/SO$_2$ molar ratios and the trend was further superimposed for half of the time interval by a periodic pattern with a 13.1 days periodicity. Therefore the periodic pattern in the BrO/SO$_2$ data may be partially explained by

the variations in the relative humidity, with a cross-correlation coefficient of up to 33 %. Also a local repetition of independent one-time events can not be ruled out from the list of possible origins, however, the probability for those to occur equidistantly in time all 13.7 days is presumed to be low. Therefore the Earth tidal forcing appears to be the most promising explanation for the observed periodical $BrO/SO_2$ variations, where correlation coefficients of 36 % and 47 % were found between the $BrO/SO_2$ molar ratios and the vertical and North-South components of the tide-induced surface displacement, respectively. The observed periodicity of 13.7 days implies a mechanism which is linked to the North-South component of the tides, e.g. an amplification of the oscillation by a North-South displacement of the host rock, rather than a mechanism which is linked to the much larger tide-induced vertical displacement.

Whether or not the presented data show a manifestation of the Earth tides, or are due to some other cause is yet difficult to answer as we only observed six consecutive periods. Nevertheless, the NOVAC dataset allows to conduct large-scale statistical analyses on this topic. Thus, long time-series at a particular volcano might give a deeper insight in the origin of such signals. Future studies will undertake a global comparison of different volcanoes and might reveal which geological geometries are excitable by the tidal forcing.

## Appendix A: Theoretical background on Earth tides

### A1 Tidal harmonics

Gravitational forcing caused by the Moon and the Sun in their interaction with the Earth varies with time. The resulting tidal potential (i.e. the residual gravitational potential) is a function of location and time and depends on all astronomical motions such as orbit, rotation, and precession of the involved bodies. The periodicities of these parameters have been studied since millenia and are well known. Already in the 1880s the tidal potential could be calculated to high precision by a harmonic decomposition (Darwin, 1883; Doodson, 1921), i.e. as a sum of an infinite number of harmonic oscillators with accurately known angular frequencies $\omega_i$ and amplitudes $a_i$. In the literature, the amplitudes of the tidal harmonics are conventionally noted as the (theoretical) vertical displacement $\Delta z_i = \frac{a_i}{g_{std}} \cdot R_{eq}$ of the Earth's surface at the equator, with respect to the equatorial Earth radius $R_{eq} = 6378\,\text{km}$ and the conventional standard gravitational acceleration $g_{std} = 9.80665 \frac{m}{s^2}$, caused by the forcing of the particular tidal harmonic (Agnew, 2007). The tidal harmonics with the largest displacement amplitudes are summarised in Table A1. Hereby, the strength of a particular tidal harmonics varies with latitude: The semi-diurnal tides are maximum at the equator and the poles but vanish at mid-latitudes. The diurnal tides are maximum at the mid-latitudes but vanish at the equator and poles. The long-term tides are maximum at the equator and vanish at the poles.

### A2 Tidal potential at the equator (at Cotopaxi)

The diurnal tidal harmonics are not contributing to the tidal potential at the equator. The strongest tidal long-term pattern are therefore caused by the interferences of the semi-diurnal tidal harmonics. The rotation of the Earth gives rise to the principal lunar semi-diurnal tide M2 and the principal solar semi-diurnal tide S2. These two strongest tidal harmonics beat together

**Table A1.** Strongest tidal harmonics (Agnew, 2007). The tide species describes its dependency on latitude (see text for details). The Darwin Symbols are a set of conventional notation, the amplitude gives the theoretical amplitude of the periodical vertical displacement of the Earth's surface which would be caused by this particular tidal harmonic alone. The frequency is given in cycles per day and the length of the period is given in days. The set is divided by the horizontal bars in semi-diurnal, diurnal, and long-term harmonics.

| Tide species | Darwin Symbol | Amplitude [m] | Frequency [cpd] | Length of Period [d] |
|---|---|---|---|---|
| | M2 | 0.63221 | 1.9322736 | 0.5175251 |
| semi-diurnal | S2 | 0.29411 | 2.0000000 | 0.5000000 |
| tides | N2 | 0.12105 | 1.8959820 | 0.5274312 |
| | K2 | 0.07991 | 2.0054758 | 0.4986348 |
| | K1 | 0.36864 | 1.0027379 | 0.9972696 |
| diurnal tides | O1 | 0.26223 | 0.9295357 | 1.075806 |
| | S1 | 0.12199 | 0.9972621 | 1.002745 |
| long-term | Mf | 0.06661 | 0.0732022 | 13.66079 |
| tides | Mm | 0.03518 | 0.0362916 | 27.55459 |
| | Ssa | 0.03099 | 0.0054758 | 182.6217 |

every 14.77 days, a periodic pattern which is commonly known as "spring-neap tide cycle". The elliptic shape of the Moon's orbit causes a modulation of the spring-neap tide cycle with additional beats every 27.55 days (N2+M2) and every 9.61 days (N2+S2), resulting in an alternation of the spring tidal amplitude (see Figure 1). The inclination of the Earth's orbit with respect to the ecliptic (contribution of the lunisolar semi-diurnal tide K2) entails a further modulation of the tidal signal with additional

5  beats every 13.66 days (K2+M2), every 182.62 days (K2+S2), and every 9.13 days (K2+N2). The most obvious impact of K2 is the semi-annual modulation (K2+S2) of the spring-neap tide cycle. The beat periodicities of all those interferences of semi-diurnal tidal harmonics are matched by long-term tidal harmonics with identical periodicities, e.g. (M2+K2) and the lunisolar fortnightly tide Mf. The beats and the long-term harmonics interfere, however, except for Mf the amplitudes of the long-term harmonics are rather small compared to the beats of the semi-diurnal harmonics.

The tidal potential results in a displacement of the Earth's ground surface in all three spatial directions. At the equator, the tide-induced displacement is strongest in the vertical component, weaker in the East-West component, and weakest in the North-South component (see Figure 1 lower panel). The vertical and the East-West component of the tide-induced displacement follows the modulated spring-neap tide cycle. In contrast, the North-South component follows a rather irregular pattern

15  approaching maximum displacement rates roughly every 13-14 days. The discrepancy in beat rate can be explained by the relative impact of the tidal harmonics exerting differential crustal strains in different spatial dimensions. In particular, if the Earth would not be tilted with respect to the ecliptic, there would be no displacement in North-South direction at the equator. Accordingly, the North-South displacement is more sensitive for those tidal harmonics which contribute due to the inclination

of the Earth, which is primarily K2. In this reasoning, the tidal harmonics with the strongest North-South component are K2 and M2, thus the dominant long-term pattern has a beat of 13.66 days (K2+M2), rather than the spring-neap tide cycle. The irregularities in the beat rate of the North-South component may be manifestations of the interferences (N2+M2), (N2+S2), and (N2+K2). Furthermore, the North-South component also follows the semi-annual modulation (K2+S2).

The compartments of the Earth respond differently to the temporal changes of tidal potential according to their mechanical properties such as their viscosity and compressibility. Solid rock is displaced by about 0.2980 (radial Love number of the SNREI Earth, see Agnew, 2007) times the theoretical value, e.g. $\pm 0.3$ m at spring tide. In contrast, magmatic melt is a fluid with a higher compressibility than solid rock and may therefore adopt stronger to the tidal potential. Accordingly, the tidal

forcing may lead to a relative displacement between magma in a conduit and the surrounding host rock.

## Appendix B: Experimental Data: Methods and Measurement

### B1    DOAS and NOVAC

Differential Optical Absorption Spectroscopy (DOAS) (Platt and Stutz, 2008) is an analysing technique which is suitable for remote sensing of volcanic gas plumes. The output of DOAS are the simultaneously retrieved slant column densities (SCD) of

a set of volcanic trace gases. The SCDs are the integral of the particle density of the particular gas along the light path. Together with additional information on the expansion of a volcanic gas plume and the wind conditions, the particle density and even the emission fluxes of volcanic trace gases like $SO_2$ and BrO can be retrieved from the SCDs. In this manuscript, we are focusing on the BrO/$SO_2$ molar ratio in volcanic gas plumes, i.e. non of those additional information is required. At volcanoes, usually scanning DOAS systems are used because these can provide a spatial coverage from horizon to horizon with a good temporal

resolution. The instruments in the Network for Observation of Volcanic and Atmospheric Change (NOVAC) were designed in an autarkic and simple set-up in order to match the harsh conditions and remote locations of volcanoes (Galle et al., 2010). As a drawback, NOVAC instruments do not have an active temperature control as the power supply by solar panels is limited. Prior to each scan ($\sim 5 - 15$ min), the measurement routine adjusts the exposure time to the meteorological conditions. As the main goal of NOVAC is the monitoring of volcanic $SO_2$ fluxes, the number of consecutive exposures is manually fixed aiming for the

best compromise between precision and temporal resolution of the $SO_2$ measurement. While this leads typically to a precision of the $SO_2$ retrieval in the percent range, the setting is not optimal for the BrO retrieval. Nonetheless, Lübcke et al. (2014) have shown that also BrO can be retrieved from NOVAC instruments by applying an advanced evaluation algorithm. Thus NOVAC allows, for the first time, to determine long-term time series of BrO/$SO_2$ molar ratios at a large number of volcanoes.

### B2    Retrieval of BrO/$SO_2$ molar ratios from NOVAC data: Plume detection and optimising the BrO detection limit

For all spectroscopic retrievals of $SO_2$ and BrO, we applied the DOAS fit scenarios described by Lübcke et al. (2014) and also most evaluation steps were applied identically. In the first evaluation step, all spectra from a horizon-to-horizon scan are

pre-evaluated for $SO_2$ using its particular zenith spectrum as reference spectrum. The result is a $SO_2$-SCD distribution as a function of the viewing angle. While the precision of the $SO_2$ data allows for a spatial analysis of the volcanic plume, the optical density of BrO in a volcanic gas plume is at least one order of magnitude smaller than for $SO_2$ and thus a higher photon number contribution is required in order to enhance the signal-to-noise ratio. This is realised by a subsequent adding of spectra
which are recorded in the temporal proximity and in the same or at least similar viewing direction.

Accordingly, we used the $SO_2$-SCD distribution to identify all spectra which are predominately part of the volcanic plume, i.e. with only little dilution, and add the spectra in order to get one "added plume spectrum per scan". The drawback of this method is the loss of spatial information as the retrieval gives only one mean plume value for the BrO-SCDs or the BrO/$SO_2$,
respectively. A volcanic gas plume can be assumed to have an approximately Gaussian shaped angular gas distribution embedded in a flat, gas free reference region. (The reference region might be differing from zero $SO_2$ as the arbitrarily picked reference spectrum might contain gas. However, at this stage only the angle-dependent shape and not the quantitative value is of interest.)

For their proof of concept, Lübcke et al. (2014) ignored this reasonable assumption of the plume shape and simply defined
the plume region as the angle region ($\geq 33°$) with the highest running mean value over 10 spectra for the $SO_2$-SCDs. For relatively narrow volcanic plumes this fixed plume region may contain also viewing directions where the volcanic plume has been significantly diluted, and therefore the "added plume spectrum" would contain also a significant contribution from the reference region and thus underestimate the $SO_2$-SCD in the real plume. To improve the accuracy, we instead fit a Gaussian distribution on the angular $SO_2$-SCD distribution in order to routinely separate all pure plume spectra from significantly diluted
spectra. The 1-$\sigma$ range of the Gaussian distribution is then defined as the plume region. When no or only a partial plume is detected (concretely, if the Gaussian fit does not succeed or when the 1-$\sigma$ range of the Gaussian fit was less then $5°$) the scan is rejected from the further analysis. On the other hand, when the 1-$\sigma$ range is expanding more than $33°$ (67 % of scans for the presented data), the plume region is defined as by Lübcke et al. (2014). Similarly and in accordance with Lübcke et al. (2014), the 10 spectra of the scan with the lowest $SO_2$-SCDs are considered as gas-free background spectra and are added up to one
"added reference spectrum per scan".

In the next step, for each scan a $SO_2$ and a BrO DOAS-fit has been performed based on the added plume spectrum and added reference spectrum. In order to further improve the signal-to-noise of the BrO fits, the added plume spectra of four consecutive scans are added up to "multi-scan plume spectra", however, only those scans with $\chi^2$(BrO fit) $< 10^{-3}$ ($\approx 95\%$ of
all suitable scans) were considered in order to avoid the influence of relatively bad scans. Analogue, also multi-scan reference spectra were obtained. The final BrO data have for periods with significant BrO degassing a mean temporal resolution of 10 (and up to $\sim 30$) data points per day and a standard deviation of about $3 \cdot 10^{13} \frac{molec}{cm^2}$. The standard deviation is estimated as two times the DOAS fit error of the multi-scan BrO fit (Stutz and Platt, 1996). The $SO_2$ detection limit of the final data is in the order of $1 \cdot 10^{16} \frac{molec}{cm^2}$ (displayed by the dashed line in the uppermost panel of Figure 3), i.e. the precision of the $SO_2$ data is still
about a factor of 100 better than for BrO and thus within the scope of BrO/$SO_2$ molar ratio calculation considered as exactly

measured.

The BrO detection limit can be further enhanced by e.g. daily averaging to dynamic values of $\mathrm{BrO}^{DT} = \frac{3}{\sqrt{n}} \cdot 10^{13} \frac{\mathrm{molec}}{\mathrm{cm}^2} \leq 1 \cdot 10^{13} \frac{\mathrm{molec}}{\mathrm{cm}^2}$ (with $n$ the number of "multi-scan" data points per day), but anyway, most retrieved BrO-SCDs are below the de-
tection limit (displayed by the dashed line in the second panel of Figure 3, and empty circles in third upper panel of Figure 3). However, a blindfold rejection of all data with BrO-SCDs below the detection limit would not only drastically reduce the number of available data but would also select those scans when the BrO-SCDs is particularly high. As such a selection leads to systematically overestimated molar ratios, a more careful selection criteria is required for more accurate BrO/SO$_2$ molar ratio calculation (Lübcke, 2014). As we are actually not interested in the BrO-SCDs but the BrO/SO$_2$ molar ratios, we set our criteria
in order to increase the reliability of observed (low) molar ratios. For this purpose we define periods of strong degassing as above an SO$_2$-SCDs threshold level of $\mathrm{SO}_2^{thres} = 7 \cdot 10^{17} \frac{\mathrm{molec}}{\mathrm{cm}^2}$ and select only those data which are above the threshold. Within those data, the maximum detection limit of the BrO/SO$_2$ molar ratios is $(\mathrm{BrO/SO}_2)^{DT} = \frac{\mathrm{BrO}^{DT}}{\mathrm{SO}_2^{thres}} = \frac{4}{\sqrt{n}} \cdot 10^{-5} \leq 1 \cdot 10^{-5}$. The SO$_2$ threshold is chosen as best compromise between a low BrO/SO$_2$ detection limit and sufficient amount of data passing the criteria. For observed SO$_2$-SCDs further above the threshold also the BrO/SO$_2$ detection limit further decreases. As conse-
quence, also some BrO/SO$_2$ data which fluctuate around zero can pass the selection criteria (i.e. whose BrO-SCD is below the BrO detection limit). For the preceding reasoning, we interpret those data (or at least there long-term averages) as reliable observations of rather low BrO/SO$_2$ molar ratios. In the third upper panel of Figure 3, we highlighted data above or below the BrO detection limit by full and empty circles, respectively. However, both kind of data were treated identically in the subsequent analysis steps.

**B3   Alternative SO$_2$ fit range for untypically strong SO$_2$ emissions**

The approximations which are applied in the DOAS algorithm are only justified for weak absorbers, i.e. if the investigated optical density is below $\sim 0.1$. For a larger optical density, the DOAS approach usually underestimates the SCD of the investigated trace gas. For the "standard" (i.e. following Lübcke et al., 2014) SO$_2$ fit range from 314.8 - 326.8 nm, SO$_2$ can be considered to be a weak absorber for SCDs of up to $\sim 1 \cdot 10^{18} \frac{\mathrm{molec}}{\mathrm{cm}^2}$ but the SO$_2$-SCD gets increasingly underestimated with increasing optical
densities. For most NOVAC data, the SO$_2$-SCDs are not exceeding this threshold significantly, however, during the Cotopaxi unrest 2015 SO$_2$-SCDs of up to the order of $1 \cdot 10^{19} \frac{\mathrm{molec}}{\mathrm{cm}^2}$ have been observed. For more accurate SO$_2$-SCDs, we re-evaluated the "multi-scan plume spectra" with an alternative fit range from 326.5 - 335.3 nm which has been suggested by Hörmann et al. (2013) in order to evaluate SO$_2$-SCDs up to about $1 \cdot 10^{19} \frac{\mathrm{molec}}{\mathrm{cm}^2}$. As drawback, the SO$_2$ measurement error is here drastically increased with relative values of about 10 %, i.e. about $1 \cdot 10^{18} \frac{\mathrm{molec}}{\mathrm{cm}^2}$ for the extreme SCDs. In order to pick the most reliable
data, we use by default the results of the "standard" SO$_2$-fit but switch to the "alternative" fit if the discrepancy between both fit results is larger than four times the fit error of the alternative fit (thus about two times the measurement error of the SO$_2$-SCD, Stutz and Platt, 1996), i.e. if the discrepancy is more than about $2 \cdot 10^{18} \frac{\mathrm{molec}}{\mathrm{cm}^2}$.

## B4 Challenges when applying the oxygen dimer O$_4$ and Colour Index retrieval on NOVAC data

Established cloud detection algorithms base on the absorption of the oxygen dimer O$_4$ at 360 nm or the relative intensity ratio of the intensity of the backscattered solar radiation at 320 nm and 440 nm (see e.g. Wagner et al., 2014). The application of those or similar algorithms on NOVAC data faces at least three challenges: (1) The effectively detected wavelengths range from
280 nm to 400 nm, however, with a linear decrease in transmittance already from 370 nm to 400 nm due to the HOYA U330 UV bandpass filter. Accordingly, the standard colour index pair is not applicable. (2) The measurement routine automatically optimises the integration time of the spectra with respect to the wavelength range of the SO$_2$ retrieval ($\sim 310 - 335$ nm). As drawback, spectra are often, in particular at cloud free days, over-saturated starting a 365 nm but sometimes even already at 359 nm. Accordingly, the standard O$_4$ retrieval is typically not applicable on NOVAC data when the sky is cloud free. On the
other hand, the absorption signal of the O$_4$ line at 344 nm is not significant for a spatial analysis. (3) Many NOVAC instruments (at Cotopaxi three of four) use a conical scan geometry thus never record zenith spectra. Missing the zenith value for O$_4$ or the colour index makes the interpretation of the analysis results more challenging.

We anyway tested both methods on the NOVAC data retrieved from the Cami station (because there zenith information is available). For the colour index analysis, we compared the mean intensity of the wavelength ranges at 329-329.7 nm and at
381-381.6 nm (each the average of 10 detector channels, see also Lübcke, 2014). We applied the analysis on the zenith spectra of each particular scan. The resulting time series of the colour index has not shown any periodic pattern and most long-term variations were small compared to the mean daily variations (see Figure A1 upper panel). For the O$_4$ retrieval, we applied a fit scenario with the same absorbers as for BrO (see Lübcke et al., 2014) but with a fit range from 338 nm to 365 nm. we applied the analysis on the individual spectra within each scan and used the particular zenith spectrum as reference spectrum.
A comparison with the results of the SO$_2$ retrieval showed no correlation, i.e. no systematic enhancement or reduction of O$_4$ absorption signal has been observed within the volcanic plume.

As additional approach, we applied the same O$_4$ analysis on the multi-scan plume spectra recorded by all NOVAC stations at Cotopaxi and used the particular multi-scan reference spectrum as reference spectrum. The resulting time series neither contained periodic patterns nor showed a significant correlation with the BrO/SO$_2$. On the other hand, the stations Refugio and
Refugio Sur, where the plume is typically passing at lower elevation angles, found O$_4$ enhancements by about $2 \cdot 10^{42} \frac{\text{molec}}{\text{cm}^2}$ in the plume while the stations Cami and San Joaquin, where the plume is typically passing in about zenith direction, found no O$_4$ enhancement in the plume (see Figure A1 lower panel). This enhancement is probably solely due to the dependency of the background O$_4$ slant column density on the elevation angle, illustrating the difficulty to interpret the O$_4$ results. We want to highlight that more sophisticated and well validated studies which e.g. take use of massive filtering of the data may
indeed retrieve information of the aerosol load from the NOVAC data, however, the development of such a sophisticated cloud classification algorithm is beyond the scope of this study.

*Acknowledgements.* The authors thank Andrea Di Muro and two anonymous reviewers for their comments on the manuscript, which greatly helped to improve it. We would like to thank the European Commission Framework 6 Research Program for funding of the NOVAC project.

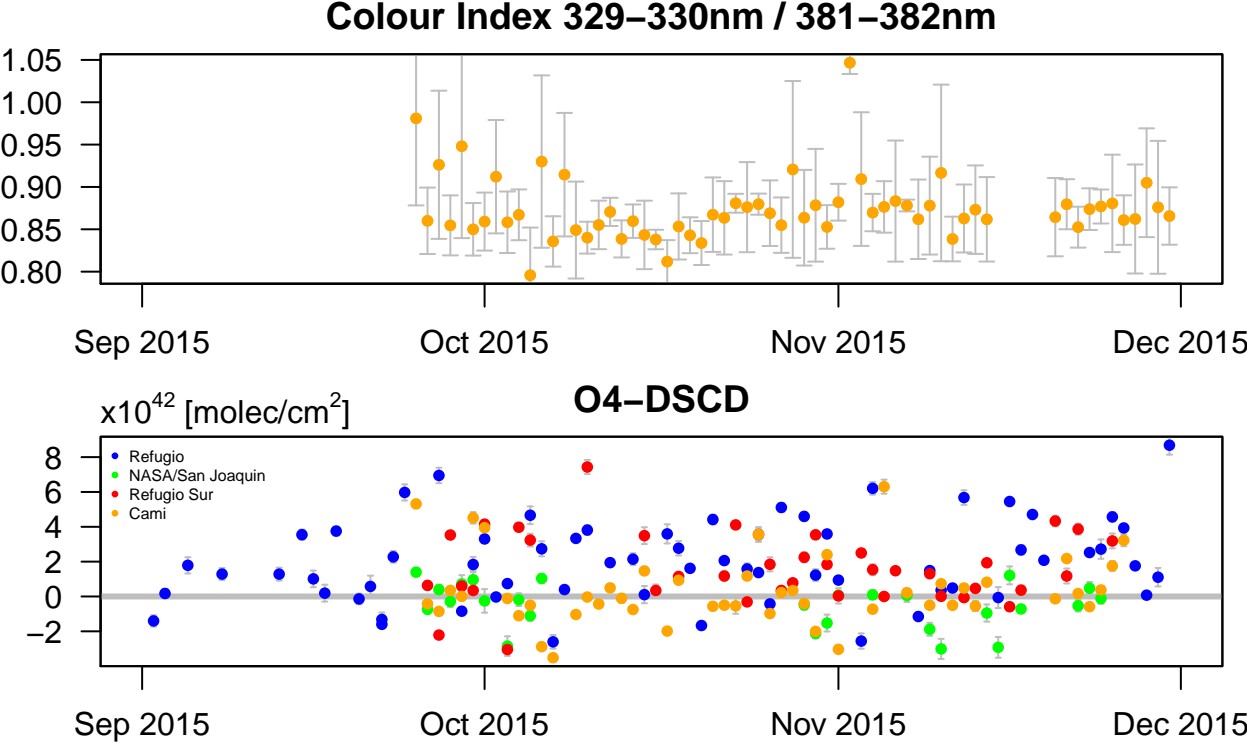

**Figure A1.** Upper panel: Colour Index (here: mean intensity of the wavelength ranges at 329-329.7 nm and at 381-381.6 nm) retrieved from Cami station. The analysis based on the single scans, shown are the daily averages and the error bars indicate the standard deviations of the daily variations. Lower panel: Different of the $O_4$ slant column density in the plume region and the reference region (as found from the $SO_2$ analysis, see Appendix B2). The $O_4$ enhancement observed by the stations Refugio and Refugio Sur is probably just due to the dependency of the background $O_4$ slant column density on the elevation angle

We kindly acknowledge the staff at IGEPN for keeping the instruments running at Cotopaxi. We thank the Deutsche Forschungsgemeinschaft for supporting this work within the project DFG PL193/14-1. Nicole Bobrowski thanks for financial support from DFG BO 3611/1-2 and the VAMOS project. Florian Dinger thanks ECMWF for providing the meteorological data and Steffen Dörner (MPIC) for assisting with the ECMWF data.

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
