# Peer review of "Periodicity in the $BrO/SO_2$ molar ratios in the volcanic gas plume of Cotopaxi and its correlation with the Earth tides during the eruption in 2015"

_Solid Earth, 2017_

## Referee Comment (RC1) · Anonymous Referee #1 · 2 Nov 2017

The discovery of a 14 day cycle in the BrO/SO2 ratio is very interesting and has the potential to shed some light into mechanisms that link the outgassing to triggering phenomena. The introduction outlining the history of DOAS measurement is also interesting, but given its length and weight, is slighltly beside the point. The interpretation of the volatile ratio in terms of the tidal potential, however, is somewhat naive, and in places strictly wrong.

- The semi diurnal peak-to-peak modulation between S2(K2) and M2 is depicted in Fig1 and has a period of 14.7 days as the excitation mechanism, while the periodogram in

Fig 3 reveals a period of 13.7 days describing the response. What is your take on the difference.

- The statement " The North-South component of the tide has no unique periodicity but a mean periodicity of 13-14 days" in Fig1 reveals the partial understanding of the authors about the tidal potential; this general statement should be removed.

- Other peaks in the periodigram in Fig 3 are attributed to "probable just artifacts due to spectral leakage" without any further comment. Spectral leakage is caused by the taper length of the time window, and could have been properly defined, if it is indeed the reason for the additional peaks.

- Fig 5: the expectation of a phase shift between excitation and response is indeed justified and could provide important information about the underlying mechanism. In this way sediment porosity, e.g., has been determined by evaluating the response of water-filled boreholes to the tidal potential. After applying a phase shift of about 1.7 and 10 days, respectively, the resulting correlation between tides and volatile ratio is merely 0.47, which is not convincing at all. Fig 6 (left panel) demonstrates the weak significance. In the conclusions the authors describe the correlation with humidity as only 33% while 36% is considered a promising explanation??

- Appendix A: Addressing the response of the Earth to the tides "The water in the oceans responds..." the authors seem to confuse the amplitude response with the phase. Ocean tides can be completely out of phase with the body tides due to eigen oscillations in bays and estuaries, while the response of solid rock in the crust is smaller than 1 degree, because it is elastic. Love numbers describe exactly this effect. Hence, the final conclusion about the relative displacement between melt and elastic rock needs to be re-considered in that light.

Hence, the data set is certainly worth investigating, but the tidal analysis presented so far needs mayor improvements.

---

## Referee Comment (RC2) · Anonymous Referee #2 · 12 Nov 2017

Reviewer's comments for:

**Periodicity in the BrO/SO$_2$ molar ratios in the volcanic gas plume of Cotopaxi and its correlation with the Earth tides during the eruption in 2015**

Florian Dinger, Nicole Bobrowski, Simon Warnach, Stefan Bredemeyer, Silvana Hidalgo, Santiago Arellano, Bo Galle, Ulrich Platt, and Thomas Wagner

Solid Earth, 2017-89

This paper by Dinger *et al.* entitled "Periodicity in the BrO/SO$_2$ molar ratios in the volcanic gas plume of Cotopaxi and its correlation with the Earth tides during the eruption in 2015" is an interesting study on periodic patterns of the BrO/SO$_2$ molar ratios in the volcanic plume from Cotopaxi volcano and their correlations to surface displacements induced by the Earth tides and to meteorological conditions. They analyze the time-series of BrO/SO$_2$ molar ratios in the volcanic plumes from Cotopaxi volcano and found that the ratio had a period of 13.7 days. They compare the period to the Earth tides and meteorological conditions. Finally, they found a good correlation between the BrO/SO$_2$ molar ratio and the N-S components of the ground displacement induced by the Earth tides and between the ratio and the relative humidity. They suggest that this correlation would be related to "excitation" of the magmatic system by the earth tides and list possible volcanological origins. The methods and the data are clearly presented and the results are convincing. The conclusions contribute to our understanding of periodic patterns of the volcanic gas emissions and the possibility of the impact of the Earth tides on the magmatic system. This article will be of interest not only to geochemists of volcanic gas emissions but also to volcanic geophysicists and other geophysicists in the surrounding disciplines. This paper is valuable to be published in this journal but needs some minor revisions of the following comments.

**Major comments**

1. This paper discusses the relation of BrO/SO$_2$ ratios and the displacements of host rocks or other magmatic systems induced by the Earth tides in Section 5.2.3 and Section 5.3. I cannot understand the image of the ground motions in the volcano that can influence the volcanic gas emissions especially ratios. The discussions about the Earth tide are divided into two sections in Discussion and it is difficult for readers to get the key points and the situations you mentioned. Please make it clear about the image of Earth-tide-induced displacements that can occur in the volcanic and magmatic systems, and then discuss about the situation of magmatic systems of Cotopaxi volcano. The influence of the Earth tides on the ratio of volcanic gas is also unclear. Please mention about it in Discussions. Here I list some papers about Earth tides and the volcanic activity that you did not cite: Sottilli et al., 2007, Effects of tidal stress on volcanic activity at Mount Etna, Italy, GRL; Sottilli and Palladino, 2012, Tidal modulation of eruptive activity at open-vent volcanoes: evidence from Stromboli, Italy, Terra Nova.

2. About the magmatic system of Cotopaxi volcano, you only mentioned about the recent unrest and the plug formation in the conduit in Section 2.3. To discuss about the magmatic system in Discussion, I think you need to give us some information about the location and depth of the magma chamber. Are there any previous studies of geophysics (distributions of hypocenters, location and geometry of source of the ground deformation) on Cotopaxi volcano? If so, you should cite such kind of papers. And it would be better to write the details about the plug formation and its reliability.

3. In Section 3.2, you did not mention about the effect of volcanic ash in retrieval of the $SO_2$ and BrO column amounts. There is some dilution effect in scanning DOAS systems (e.g., Mori et al., 2006, GRL; Kern et al., 2010, BV) and the existence of volcanic ash can result in underestimation of the column amounts. Please mentioned about the problem in this section.

**Minor comments**

4. In general, the order of appearance of the figures does not match to the order of reference of the figures in the text. This can be confusing for the readers.

5. Page 1, line 6: "One strong aspirant …" is in plural form. It should be in singular form.

6. Page 2, line 6: "… (COSPEC, M. M. Millan (1970))" Is it correct using parentheses in parentheses? The other citations have similar problems, so please check the style of the journal.

7. Page 3, line 27: "olique" is "oblique", isn't it?

8. Page 4, line 30: "… and composition (Gaunt et al., 2016) of ash emissions …" should be "… and composition of the emitted ash (Gaunt et al., 2016) …"

9. Page 9, line 6: "Figure 6" is Figure 5?

10. Page 10, line 26: "contrast, the O4 analysis requires …" O4 should be $O_4$.

11. Page 16, line 6: "the Bro/$SO_2$ data …" should be "the BrO/$SO_2$ data …".

---

## Author Comment (AC1) · 9 Dec 2017

**Reply on Anonymous Reviewer No. 2 (AR#2)**

We thank the reviewer for acknowledging the relevance of our gas data.

**(1a) AR#2: The discussions about the Earth tide are divided into two sections in Discussion and it is difficult for readers to get the key points and the situations you mentioned. Please make it clear about the image of Earth-tide-induced displacements that can occur in the volcanic and magmatic systems, and then discuss about the situation of magmatic systems of Cotopaxi volcano.**

**Change:** We collected the discussion on the Earth tides in a new section 5.4 in order to improve the readability. Further, we switched the order of the remaining section 5.2 and 5.3 for better readability.

**(1b) AR#2: The influence of the Earth tides on the ratio of volcanic gas is also unclear. Please mention about it in Discussions.**

**Change:** We added the following clarification to section 5.4:
*"(...) In this reasoning, the shallow emplacement of magma prior to the phreatomagmatic explosion in August 2015 may have (temporarily?) shifted the geometry of the Cotopaxi magma plumbing system from a non-excitable to an excitable state. Additionally, the explosive activity very likely gave rise to a transition from closed system to open system degassing, and thus made the volcano more perceptive to external influences. Further, combining our results and the interpretation from Gaunt et al. (2016) suggests a possible tide-induced repetitive plug formation and destruction, causing an alternation of "open and almost shut condition" as proposed by Fischer et al. (2002) for explosive activity at Karymsky volcano. All those effects can result in a periodic variation of the pressure regime in the shallow magmatic system, leading to a periodic variation of the volatile solubility in the magmatic melt, which in turn may vary the magnitude and/or composition of volcanic gas emissions."*

**(1c) AR#2: Here I list some papers about Earth tides and the volcanic activity that you did not cite:...**

We thank the reviewer for providing this supporting literature!

**Change:** We use those literature for clarifying our interpretation (see section 5.4).

**(2a) AR#2: To discuss about the magmatic system in Discussion, I think you need to give us some information about the location and depth of the magma chamber. Are there any previous studies of geophysics (distributions of hypocenters, location and geometry of source of the ground deformation) on Cotopaxi volcano? If so, you should cite such kind of papers.**

**Change:** We added to section 2.1 *"Observations of ground deformation and hypocenter distributions of volcanic earthquakes made in 2001/2002 (Hickey et al., 2015) and 2015 (Morales Rivera et al., 2017) suggest that Cotopaxi currently has a shallow magma reservoir beneath the southwestern flank, which is located at a depth of approximately 5-12 km below the summit."*

**(2b) AR#2: And it would be better to write the details about the plug formation and its reliability.**

The possibility of a repetitive plug formation and destruction was proposed by Gaunt et al., 2016, as a possible interpretation of their "enigmatic" ash data. We think this was rather a plausible speculation than a empirical evidence and do not want to discuss the reliability. Nevertheless, we think this possible co-occurrence should be included to the list of possible causes.

**Change:** (none)

**(3) AR#2: In Section 3.2, you did not mention about the effect of volcanic ash in retrieval of the SO2 and BrO column amounts. There is some dilution effect in scanning DOAS systems (e.g., Mori et al., 2006, GRL; Kern et al., 2010, BV) and the existence of volcanic ash can result in underestimation of the column amounts. Please mentioned about the problem in this section.**

This effect was already discussed in the introduction, but there is no harm to highlight this issue again in the section 3.2. Further, we added the debate on ash scavenging.

**Change:** We added in section 3.2 *"From August 2015 to February 2016, Cotopaxi also emitted large amounts of ash. These ash emissions can alter the atmospheric radiative transport and thus result in an underestimation of the retrieved SCDs (e.g., Mori et al., 2006; Kern et al., 2010). Nevertheless, those underestimations are approximately the same for both gas species thus their ratio is almost not affected by those ash emissions (Lübcke et al., 2014). Further, it is under debate whether ash has a differential scavenging effect on sulphur and halogens, respectively (Bagnato et al., 2013; Delmelle et al., 2014)."*

**(4) AR#2: Minor changes**

**Change:** All applied as proposed.

**(5) Further changes**

The conclusions focused on the observation and interpretation of the periodic signal. However, also the trend of the $BrO/SO_2$ molar ratios is an important result of the observations. This trend has been discussed in the manuscript but we did not include it in the conclusions. We added also those findings to the conclusions:

*"Previous studies on the volcanic gas plumes of several volcanoes (Mt. Etna, Nevado del Ruiz, Tungurahua) observed relatively low $BrO/SO_2$ molar ratios prior to volcanic explosions and an increasing trend in $BrO/SO_2$ molar ratios afterwards. Those consistent observations raised the question whether the $BrO/SO_2$ molar ratios can be interpreted as a precursor of volcanic activity. We observed a similar behaviour at Cotopaxi during its unrest period in 2015, extending the empirical foundation of this claim. At Cotopaxi, the $BrO/SO_2$ molar ratios were almost vanishing prior to the phreatomagmatic explosions in August 2015, significantly higher after the explosions, and further increased from September 2015 to December 2015. After December 2015, the unrest calmed down accompanied by a decrease in $SO_2$-SCDs to a level lower than prior to the explosions, however, the BrO-SCDs remained relatively large. The latter observation suggests that bromine degassed at Cotopaxi predominately after sulphur from the magmatic melt."*

Addition to the abstract:
*"The $BrO/SO_2$ molar ratios were very small prior to the phreatomagmatic explosions in August 2015, significantly higher after the explosions, and continuously increasing until the end of the unrest period in December 2015. These observations together with similar findings in previous studies at other volcanoes (Mt. Etna, Nevado del Ruiz, Tungurahua) suggest a possible link between a drop in $BrO/SO_2$ and a future explosion."*

---

## Author Comment (AC2) · 9 Dec 2017

**Reply on Anonymous Reviewer No. 1 (AR#1)**

We thank the reviewer for the thoughtful and encouraging comments and for acknowledging the relevance of our gas data. The reviewer, first, highlights some misleading statements on the Earth tides themselves, second, and then goes on to question the reliability of the correlation results, and third, asks about a possible explanation for the discrepancy between the observed period of 13.7 days and the astronomic forcing with a period of 14.8 days.

We mostly agree on the criticism of the singular misleading statements due to bad phrasing in our former version and we therefore corrected them (see below). However, we do not share the reviewer's pessimistic view on a correlation coefficient of 47 %. Figure 6 illustrates clearly a sound correlation between the gas data and tidal data. Finally, we have shown that the period in the gas data of 13.7 days matches the temporal variations of the (modelled) North-South component of the tide-induced surface displacement (rather than the periodicity of the total/vertical tidal forcing). It was not the aim of this paper to identify the cause for this partial correlation. Nevertheless, the last ten lines of the discussion discuss the plausibility of a causal relationship between the correlated data, based on the specific environment of Cotopaxi. We just wanted to offer some possible directions for future investigations. Below we give our detailed responses to the individual points of this reviewer.

**(1) AR#1: The introduction outlining the history of DOAS measurement is also interesting, but given its length and weight, is slightly beside the point.**

We agree with the reviewer.

**Change:** We shortened the introduction by 12 lines (from 65 to 53 lines, see new manuscript), predominately cutting some of the DOAS part. The new introduction is structured to give an overall motivation for the research on volcanic degassing (10 lines), an overview of the data source we are using (13 lines), an introduction in the analysis of $BrO/SO_2$ molar ratios (17 lines), and literature on the comparison of NOVAC data and Earth tides (13 lines).

**(2) AR#1: The semi diurnal peak-to-peak modulation between S2(K2) and M2 is depicted in Fig1 and has a period of 14.7 days as the excitation mechanism, while the periodigram in Fig 3 reveals a period of 13.7 days describing the response. What is your take on the difference.**

We gave a good reason for this apparent discrepancy, which we clarify in the revised manuscript, see below.

**Change:** The interpretation of the correlation with the Earth tides was formerly split in section 5.2 and 5.3. In the revised manuscript, we collected all interpretation in the new section 5.4 in order to improve the readability of the discussion (see also reply on

the second review). There you can now find, among others, the following interpretation:

*"Tide-induced processes are intuitively expected to strictly follow the periodicity of the strongest tidal long-term pattern, that is the spring-neap tide cycle with period of 14.8 days. The observed periodic pattern of 13.7 days in our $BrO/SO_2$ data, however, matches much better with the temporal intraday amplitude variation of the North-South component of the tide-induced surface displacement, which follows a rather irregular pattern with maxima occurring roughly every 13-14 days. In other words, $BrO/SO_2$ ratios were elevated when the tidal amplitude variations in North-South direction were most pronounced. Our results accordingly suggest that the volcanic system of Cotopaxi (currently?) is more sensitive to tide-induced stresses acting in the North-South direction, rather than to stresses in the vertical and East-West directions.*
*This interpretation is further supported by the orientation of the local fault system and associated ambient stress field at Cotopaxi, which is located in a transfer fault zone with greatest principal stress acting in the ENE-WSW direction and the weakest principal stress in the North-South, i.e. also in horizontal direction (Fiorini and Tibaldi, 2012). Such a setting favours the intrusion and ascent of magma along East-West striking planar structures, which is further corroborated by the observation of the inclined sheet intrusion beneath the south-western flank of the volcano (Morales Rivera et al., 2017). Thus, the additional tide-generated stresses probably have a much higher relative impact, when they act in the direction of weakest principle stress, i.e. normal to East-West striking compressible magma pathways, if compared to the other directions. Such a directional dependency is indeed well known for the tidal response of inclined planar aquifers, which cross-cut borehole wells. Bower (1983) e.g. reported oscillations of water levels in boreholes in Canada, which indicated a strong response to the horizontal component of the semi-diurnal M2 tide acting normal to the strike direction of the intersecting aquifers."*

**(3) AR#1 The statement "The North-South component of the tide has no unique periodicity but a mean periodicity of 13-14 days" in Fig1 reveals the partial understanding of the authors about the tidal potential; this general statement should be removed.**

We argue that the content of this statement is crucial for the discussion (see (2)) but agree that "mean periodicity" is not an appropriate term.

**Change:** We replaced the statement by *"The North-South component of the tide has no strictly regular periodicity but reaches a maximum roughly every 13-14 days and is increasing from September to December 2015."*

The details of the tidal dynamics are virtually numberless and are even more complex when discussing the particular spatial components of the tidal potential. We nevertheless believe that our general understanding of the tidal dynamics is sufficient to interpret the tidal potential (for any given form) as well as its interaction with the volcanic system. Further, the analysis as presented in this paper is not affected by missing knowledge

on the ultimate origin of the pattern in the North-South component, i.e. the tidal time series are just treated as input data.

This said, we can frankly admit that we have not completely understood the ultimate origin of the irregular pattern in the North-South component. Nevertheless, we can give a plausible interpretation of the pattern in the North-South component, although we can not back it up with literature. This interpretation is now included in the manuscript.

**Change:** We revised Appendix A in order to motivate the irregular pattern in the North-South component. In particular, we added:

*"(…) the North-South component follows a rather irregular pattern with a maximum rough every 13-14 days. The discrepancy in beat rate can be explained by the relative impact of the tidal harmonics on the different spatial dimensions. In particular, if the Earth would not be tilted with respect to the ecliptic, there would be no displacement in North-South direction at the equator. Accordingly, the North-South displacement is more sensitive for those tidal harmonics which contribute due to the inclination of the Earth, which is primarily K2. In this reasoning, the tidal harmonics with the strongest North-South component are K2 and M2, thus the dominant long-term pattern has a beat of 13.66 days (K2+M2), rather than the spring-neap tide cycle. The irregularities in the beat rate of the North-South component may be manifestations of the interferences (N2+M2), (N2+S2), and (N2+K2). Furthermore, the North-South component also follows the semi-annual modulation (K2+S2)."*

**(4) AR#1: Other peaks in the periodigram in Fig 3 are attributed to "probable just artifacts due to spectral leakage" without any further comment. Spectral leakage is caused by the taper length of the time window, and could have been properly defined, if it is indeed the reason for the additional peaks.**

We are not sure whether the review refers to the common Fourier transform analysis or to the here applied Lomb-Scargle analysis (because our time series has an uneven sampling). For Lomb-Scargle, things are typically more complicated and we do not know a proper way to check/prove spectral leakage. Instead we tested the Lomb-Scargle for possible artefacts as follows: We generated a sinusoidal signal with a period of 13.7 days and with a length of 6 periods and a sampling rate of twice a day. Then we removed random data points such that we got an uneven sampling which had the same number of data points as our BrO/SO2 data. Finally, we applied the Lomb-Scargle analysis on this "gapped" sinusoidal signal.

We found that the amplitude and position of the side lobes vary for different sets of randomly removed data points. Further, the side lobes are not symmetric around the central maxima. Nevertheless, the peaks observed in Fig 3 are close to the mean amplitude and position of the set of sinusoidal test data. Thus, we can not exclude that those are just artefacts. However, we agree with the reviewer that we have not proven that those are indeed just artefacts.

**Change:** (1) We removed the sentence on the spectral leakage in order to avoid a

misleading interpretation of those minor peaks which are actually not important for the further analysis. (2) For completeness, we added the further results of the Lomb-Scargle analysis: *"On lower confidence levels, the Lomb-Scargle analysis proposes further peaks at a periodicity of 7.1 days, 9.8 days, and 18.8 days, respectively."*

**(5) AR#1: Fig 5: the expectation of a phase shift between excitation and response is indeed justified and could provide important information about the underlying mechanism. In this way sediment porosity, e.g., has been determined by evaluating the response of water-filled boreholes to the tidal potential. After applying a phase shift of about 1.7 and 10 days, respectively, the resulting correlation between tides and volatile ratio is merely 0.47, which is not convincing at all. Fig 6 (left panel) demonstrates the weak significance. In the conclusions the authors describe the correlation with humidity as only 33% while 36% is considered a promising explanation??**

In contrast to the reviewer, we consider a correlation of 47% to be unexpectedly high, convincing that gas data and North-South tide are partially correlated (see Fig 6, right panel). We see this statement justified already mathematically but also justified by Fig. 6 right panel (which is just a different graphical representation of Fig. 6 left panel), which clearly shows that there is a partial correlation between the tide-induced displacement and the gas ratio. Further, we did not aim to label correlation coefficient of 33% of the relative humidity as insignificant. Finally, we have to highlight that we have not claimed any "promising explanation" but just described the results of the statistical analysis.

We thank the reviewer for mentioning the response of water-filled boreholes. We had a similar mechanism in mind but missed the literature on the empirical studies.

**Change:** (1) We added literature on borehole observations. (2) We clarified our interpretation of the correlation results by adding the following paragraph (in the newly created section 5.4): *"The observed periodicity in the time series of the $BrO/SO_2$ molar ratios is superimposed by an increasing trend and a large scatter in the data. The latter highlights the complexity of the interpretation of $BrO/SO_2$ molar ratios, which potentially depends on an unknown number of volcanological and atmospherical mechanisms and the fluctuations of their parameters. Despite the large scattering, we nevertheless found an unexpectedly high correlation between the $BrO/SO_2$ molar ratios on the one hand and the Earth tidal forcing or the relative humidity on the other hand, with correlation coefficients of 47% and 33%, respectively. Thus, the tidal forcing as well as the relative humidity are the most probable candidates to explain a part of the variability of the $BrO/SO_2$ molar ratios. Accordingly, both mechanisms may independently contribute to the variation of the $BrO/SO_2$ molar ratios at the same time. In the following, we focus on the plausibility of a causality between the $BrO/SO_2$ molar ratios and the North-South component of the tidal forcing, which nevertheless appears to be the best candidate"*

**(6) AR#1: Appendix A: Addressing the response of the Earth to the tides "The water in the oceans responds..." the authors seem to confuse the amplitude response with the phase. Ocean tides can be completely out of phase with the body tides due to eigen oscillations in bays and estuaries, while the response of solid rock in the crust is smaller than 1 degree, because it is elastic. Love numbers describe exactly this effect. Hence, the final conclusion about the relative displacement between melt and elastic rock needs to be re-considered in that light.**

The sentence on the ocean tides has been included in order to illustrate the impact of viscosity, rather than discussing the ocean tides in detail. The reviewer is correct about the complexity of the ocean tides (anyway, our sentence was meant for the open sea). Because the ocean tides are of no relevance in the manuscript and in order to avoid misleading content, we removed the sentence on the ocean tides.

**Change:** The sentence *"The water in the oceans responds to variations of the tidal forcing almost immediately and is thus displaced always with the theoretical value, e.g. 1m at spring tide."* is removed.

In the next sentences of the manuscript, we used the terms "slower" and "faster" which are typically attributed to velocity or time (probably the reviewer's criticism focuses on those?!). In our manuscript, those should refer to the amplitude of the maximum displacement rather than a phase shift (i.e. whether the magmatic melt reach the maximum is rather a question of time than of elasticity). But the reviewer is definitively right that those terms are much more plausible when talking about a phase shift and are thus misleading here.

**Change:** We changed the two sentences to *"Solid rock is displaced by about 0.2980 (radial Love number of the SNREI Earth, see Agnew, 2007) times the theoretical value, e.g. $\pm 0.3\,m$ at spring tide. In contrast, magmatic melt is a fluid with a higher compressibility than solid rock and may therefore adopt stronger to the tidal potential."*

**(7) Further changes**

The conclusions focused on the observation and interpretation of the periodic signal. However, also the trend of the $BrO/SO_2$ molar ratios is an important result of the observations. This trend has been discussed in the manuscript but we did not include it in the conclusions. We added also those findings to the conclusions:
*"Previous studies on the volcanic gas plumes of several volcanoes (Mt. Etna, Nevado del Ruiz, Tungurahua) observed relatively low $BrO/SO_2$ molar ratios prior to volcanic explosions and an increasing trend in $BrO/SO_2$ molar ratios afterwards. Those consistent observations raised the question whether the $BrO/SO_2$ molar ratios can be interpreted as a precursor of volcanic activity. We observed a similar behaviour at Cotopaxi during its unrest period in 2015, extending the empirical foundation of this claim. At Cotopaxi, the $BrO/SO_2$ molar ratios were almost vanishing prior to the phreatomagmatic explo-*

*sions in August 2015, significantly higher after the explosions, and further increased from September 2015 to December 2015. After December 2015, the unrest calmed down accompanied by a decrease in $SO_2$-SCDs to a level lower than prior to the explosions, however, the BrO-SCDs remained relatively large. The latter observation suggests that bromine degassed at Cotopaxi predominately after sulphur from the magmatic melt."*

Addition to the abstract:
*"The $BrO/SO_2$ molar ratios were very small prior to the phreatomagmatic explosions in August 2015, significantly higher after the explosions, and continuously increasing until the end of the unrest period in December 2015. These observations together with similar findings in previous studies at other volcanoes (Mt. Etna, Nevado del Ruiz, Tungurahua) suggest a possible link between a drop in $BrO/SO_2$ and a future explosion."*

---

## Author Response (AR2)

**Reply on Editor remarks**

**(1) Editor: minor remarks**

**Change:** All minor remarks are applied as proposed or with a slight variation. Also some further suboptimal wording has been changed.

**(2a) Editor: please make a better link between the symbol in the legend (BrO $<$ detection) and the data treatment discussed in the appendix for BrO concentrations close or lower than the detection limit**

**Change:** We added to the caption of Fig. 3 *"Ratios obtained from BrO data below the detection limit of two standard deviations are highlighted by open circles (see Appendix B2)."*
And we added to the Appendix B2: *"As consequence, also some $BrO/SO_2$ data which fluctuate around zero can pass the selection criteria (i.e. whose BrO-SCD is below the BrO detection limit). For the preceding reasoning, we interpret those data (or at least there long-term averages) as reliable observations of rather low $BrO/SO_2$ molar ratios. In the third upper panel of Figure 3, we highlighted data above or below the BrO detection limit by full and empty circles, respectively. However, both kind of data were treated identically in the subsequent analysis steps."*

**(2b) Editor: In any case, I find that the definition and meaning of the long term background trend requires some clarification (...) Please clarify this point and how that affects the long term trend and the trend fitting and removal procedure as well**

The long-term trend in Fig. 3 (grey line) is only a guidance for the eye. It has been calculated with a local regression low pass filter in order to obtain a smooth but also flexible (i.e. able to follow the variations from May 2015 to May 2016) line. However, this trend and its numerical values are not used in the data analysis. We think that a further description of how this low pass filter operates is off the point.
In contrast, when subtracting the trend prior to the Lomb-Scargle analysis, this is done as already written in the text: *"Also the trend was removed by subtracting a polynomial fit to the fifth order from the $BrO/SO_2$ time series."*

**Change:** (1) We added to the caption of Fig. 3 *"The trend (grey line) is calculated by a local regression low pass filter."* (2) We added to the sentence cited above: *"(This should however not be confused with the trend shown in the third upper panel of Figure 3, which is just a smooth guidance for the eye. Nevertheless, the numerical values of those both trend fits are similar for the time interval from September to December 2015.)"*

**(2c) Editor: In 2016, reported SO2 SCD (and I assume SO2 fluxes too) were very low ($\ll$ detection limit?) like before July 2015, but calculated BrO/SO2 ratios are higher; are these estimations and differences reliable?**

The reliability of the spectroscopic data is only based on the DOAS fit quality and not on any subsequent processing (e.g. subtracting the trend). The $SO_2$-SCDs have been still exceeding the detection limit of $1 \cdot 10^{16} \frac{molec}{cm^2}$ (see Appendix B2), however, exceeded the "strong plume condition" of $7 \cdot 10^{17} \frac{molec}{cm^2}$ (see Appendix B2) only during single days.

**Change:** We added the following paragraph to the end of section 3.2: *"Around March 2016 we observed some strong gas expulsions approaching $SO_2$-SCDs, which exceeded $7 \cdot 10^{17} \frac{molec}{cm^2}$. The corresponding $BrO/SO_2$ molar ratios were about $4.4 \cdot 10^{-5}$, thus similar to the post-explosion value in September 2015. Despite that the reliability of this observation is limited due to the small sample size, these data points indicate that the long-term average value of the $BrO/SO_2$ molar ratios was about $4 - 5 \cdot 10^{-5}$ during this period."*

**(3) Editor: the figure 3 should offer a clearer link with the four reported phases of volcanic activity (only one red vertical line marks some explosions) as the time evolution of the ash emissions etc is not clearly reported**

Section 3.2 (and Figure 3) introduces and describes the time series of the gas data. The time series of the gas data could be grouped by the phases in Table 1, however those are nevertheless somewhat arbitrary. We decided to not add those semi-arbitrary grouping in Figure 3 for (1) not overloading the figure and (2) avoiding misleading interpretation of those "distinct" phases. In contrast, while there are of course similarities between ash and gas emissions, there is no obvious manifestation of the four phases reported by e.g. Gaunt et al. (2017) in the time series of the gas data. Accordingly, we did not sketch their grouping in Figure 3 in order to avoid the impression that such a grouping is identical for the gas emissions.

**Change:** (none)

**(4) Editor: I'm surprised that no general discussion is presented on the correlation (if any) between $SO_2$ flux data and the BrO/SO2 ratio.**

First of all, Fig. 5 shows that there is no correlation between $BrO/SO_2$ molar ratios and the $SO_2$-SCDs. Nevertheless, this observation was not highlighted in the text. We added a sentence on this to the text.

We do not present a comparison with $SO_2$ emission rates because it lies outside the main focus of the paper, which is to explore an observed relation between $BrO/SO_2$ molar ratios and Earth Tides. However, a comparison between $SO_2$ emission rates derived by IGEPN (Hidalgo et al., in preparation) with $BrO/SO_2$ molar ratios shows a lack of correlation between these two observables (see Figure 1 below, which presents data recorded

by San Joaquin station in September 2015). This is in general expected on physical grounds and published long-term studies (e.g. Lübcke et al., 2014 for Nevado del Ruiz) have also found a lack of correlation which indeed give value to the use of gas molar ratios as an additional parameter, independent of gas emission rate, to study volcanic activity.

[Figure]

Figure 1: Comparison of the BrO/SO$_2$ molar ratios (from this manuscript) and SO$_2$ emission rates (based on IGEPN data) for data recorded by the San Joaquin station in September 2015. The data above 100 kg/s may be artefacts of the wind model. Nevertheless even omitting those extreme values, no correlation between ratios and emission rates can be observed (correlation coefficient of 8 %).

**Change:** We added the following paragraph in section 5.1: *"The BrO/SO$_2$ molar ratios are not correlated to the daily means of the SO$_2$ slant column density (correlation coefficient of -10 %, see Figure 3) but are partially correlated to the daily means of the BrO slant column densities (39 %). It is interesting to note that similar observations can be made when comparing BrO/SO$_2$ molar ratios and SO$_2$ emission rates obtained from the same instruments. For this purpose a time series of the SO$_2$ emission rates was provided by the group at IGEPN, who have access to a novel wind model with enhanced temporal and spatial resolution allowing for the calculation of more accurate emission rates. A publication of the respective time series of the SO$_2$ emission rates is in preparation. Based on the IGEPN data, BrO/SO$_2$ molar ratios and SO$_2$ emission rates are not correlated during the period of unrest."*

However, we did not add the figure (shown in this reply) to the manuscript because (1) it adds no further content beyond the paragraph above and (2) the correlation of the overall data set will be part of the publication in preparation anyway.

[revised manuscript text omitted]
{\mathrm{molec}}{\mathrm{cm}^2}$ | $2 \cdot 10^{17} \frac{\mathrm{molec}}{\mathrm{cm}^2}$ | $\approx 0$ | $4 \cdot 10^{13} \frac{\mathrm{molec}}{\mathrm{cm}^2}$ | |
| May 22 - August 14 | $1 \cdot 10^{18} \frac{\mathrm{molec}}{\mathrm{cm}^2}$ | $2 \cdot 10^{18} \frac{\mathrm{molec}}{\mathrm{cm}^2}$ | $2 \cdot 10^{12} \frac{\mathrm{molec}}{\mathrm{cm}^2}$ | $7 \cdot 10^{13} \frac{\mathrm{molec}}{\mathrm{cm}^2}$ | $0.2 \pm 0.1 \cdot 10^{-5}$ |
| August 14 - August 22 | $2 \cdot 10^{18} \frac{\mathrm{molec}}{\mathrm{cm}^2}$ | $3 \cdot 10^{18} \frac{\mathrm{molec}}{\mathrm{cm}^2}$ | $2 \cdot 10^{13} \frac{\mathrm{molec}}{\mathrm{cm}^2}$ | $8 \cdot 10^{13} \frac{\mathrm{molec}}{\mathrm{cm}^2}$ | $1.1 \pm 0.2 \cdot 10^{-5}$ |
| August 22 - September 8 | (data gap) | | | | |
| September 8 - December 5 | $1 \cdot 10^{18} \frac{\mathrm{molec}}{\mathrm{cm}^2}$ | $1 \cdot 10^{19} \frac{\mathrm{molec}}{\mathrm{cm}^2}$ | $2 \cdot 10^{14} \frac{\mathrm{molec}}{\mathrm{cm}^2}$ | $4 \cdot 10^{14} \frac{\mathrm{molec}}{\mathrm{cm}^2}$ | $4.8 \pm 1.8 \cdot 10^{-5}$ |
| December 5 - May 1 2016 | $1 \cdot 10^{18} \frac{\mathrm{molec}}{\mathrm{cm}^2}$ | $2 \cdot 10^{18} \frac{\mathrm{molec}}{\mathrm{cm}^2}$ | $4 \cdot 10^{13} \frac{\mathrm{molec}}{\mathrm{cm}^2}$ | $1 \cdot 10^{14} \frac{\mathrm{molec}}{\mathrm{cm}^2}$ | $4.4 \pm 1.6 \cdot 10^{-5}$ |

[revised manuscript text omitted]